



# Future projections of temperature and mixing regime of European temperate lakes

Tom Shatwell[1,2], Wim Thiery[3,4], Georgiy Kirillin[1]

[1]Leibniz-Institute of Freshwater Ecology and Inland Fisheries (IGB), Department of Ecohydrology, Müggelseedamm 310, 12587 Berlin, Germany.
[2]Helmholtz-Centre for Environmental Research (UFZ), Department of Lake Research, Brückstrasse 3a, 39114 Magdeburg, Germany.
[3]ETH Zurich, Institute for Atmospheric and Climate Science, Universitaetstrasse 16, 8092 Zurich, Switzerland.
[4]Vrije Universiteit Brussel, Department of Hydrology and Hydraulic Engineering, Pleinlaan 2, 1050 Brussels, Belgium.

*Correspondence to*: Tom Shatwell (shatwell@igb-berlin.de)

**Abstract.** The physical response of lakes to climate warming is regionally variable and highly dependent on individual lake characteristics, making generalisations about their development difficult. To qualify the role of individual lake characteristics in their response to regionally homogeneous warming, we simulated temperature, ice cover and mixing in four intensively studied German lakes of varying morphology and mixing regime with a one-dimensional lake model. We forced the model with an ensemble of 12 climate projections (RCP4.5) up to 2100. The lakes were projected to warm at $0.10 – 0.11$ °C decade$^{-1}$, which is $75 – 90\%$ of the projected air temperature trend. In simulations, surface temperatures increased strongly in winter and spring, but little or not at all in summer and autumn. Mean bottom temperatures were projected to increase in all lakes, with steeper trends in winter and in shallower lakes. Modelled ice thaw and summer stratification advanced by $1.5–2.2$ and $1.4 – 1.8$ d decade$^{-1}$ respectively, whereas autumn turnover and winter freeze timing was less sensitive. The projected summer mixed layer depth was unaffected by warming but sensitive to changes in water transparency. By mid-century, the frequency of ice and stratification-free winters was projected to increase by about 20%, making ice cover rare and shifting the two deeper dimictic lakes to a predominantly monomictic regime. The polymictic lake was unlikely to become dimictic by the end of the century. A sensitivity analysis predicted that decreasing transparency would dampen the effect of warming on mean temperature but amplify its effect on stratification. However, this interaction was only predicted to occur in clear lakes, and not in the study lakes at their historical transparency. Not only lake morphology, but also mixing regime determines how heat is stored and ultimately how lakes respond to climate warming. Seasonal differences in climate warming rates are thus important and require more attention.



## 1 Introduction

Most lakes in the world tend to warm due to climate change, though the response of lakes to climate change is varying among different regions and different lake types (O'Reilly et al., 2015). Warming alters lake mixing characteristics and has consequences for lake ecology, metabolism and biogeochemistry, yet the broader impacts of lake warming remain unclear.

Recent studies highlighted that lakes respond quite individually to climate change. Although surface water temperature ($T_s$) is perhaps the most predictable indicator of warming (Adrian et al., 2009), trends in $T_s$ remain globally highly variable (O'Reilly et al., 2015). Deep water temperatures respond less predictably to warming and have been observed to increase, decrease or not change with increasing air temperature (Dokulil et al., 2006; Ficker et al., 2017; Kirillin et al., 2013; Kirillin et al., 2017; Richardson et al., 2017; Winslow et al., 2017). Stratification strength and duration generally increase due to

warming (Butcher et al., 2015; Kirillin, 2010), but patterns of change may have little regional coherence and cannot be reliably inferred from surface water trends (Read et al., 2014). We know even less about how warming influences the mixed layer depth, which is important for instance for light availability for primary production.

To better understand the sources of this variability, research has focused on identifying which characteristics modulate how lakes and particularly lake stratification respond to warming. Recently, a series of modelling studies reported variable

regional response of lakes to projected climate change in the near future (e.g. Boike et al., 2015; Butcher et al., 2015; Dibike et al., 2011; Ladwig et al., 2018; Magee and Wu, 2017; Prats et al., 2018; Woolway et al., 2017). Such regional studies typically focus on the role of lake-specific factors, such as lake size, morphometry, and water quality in long-term lake trends due to local warming. A consistent research extension in this direction could embrace upscaling to worldwide lake trends in combination with evaluating different global change scenarios and (or) different lake models. Complementary to

this "extensive" approach, lake modelling is an efficient tool for generalizing regional studies via "intensive" understanding of the energy transport between different temporal scales, which finally produce the integrated effect of slow atmospheric warming on lake dynamics. In this regard, the seasonal variations deserve special attention as representing the most energetic part of the entire spectrum in lakes as well as in the atmospheric boundary layer. High amplitudes of the seasonal variations involve the formation of density stratification and ice cover. These processes are, in turn, governed by different sets of

physical forces at different stages of the seasonal cycle, ensuring a highly non-linear lake response to atmospheric forcing on longer time scales. In freshwater lakes, the non-linear effects of seasonality get stronger as the seasonal temperature amplitudes increase relative to the maximum density value of freshwater (~4° C) and to the freezing point of 0° C. Hence, geographically, seasonality is the strongest in lakes at mid-latitudes, being damped towards tropical and polar regions.

Seasonal stratification and ice cover prevent certain water layers from interacting directly with the atmosphere for certain

periods of the year, so it follows that the mixing regime should mediate the effect of increasing air temperature on lakes. Major seasonal stratification patterns in freshwater lakes include dimictic (with lake stratification destroyed twice a year, when lake temperatures cross the maximum density value) and monomictic (with stratification destroyed once a year, when surface temperatures decrease to the maximum density value, but do not reach the freezing point, so that no stable winter



stratification forms), as well as polymictic (in lakes too shallow for a stable stratification to form) and oligomictic (in lakes too deep for stratification to be destroyed). The fact that climate warming is likely to shift the mixing regime of some lakes to the right along the polymictic – dimictic – monomictic – oligomictic continuum (Kirillin, 2010; Livingstone, 2008; Shimoda et al., 2011) makes understanding the role of mixing regime all the more important. Still, seasonal warming

patterns and their role in mediation of the long-term changes remain comparatively unexplored (Winslow et al., 2017).

Lake morphology has been identified as a key factor affecting the seasonal mixing regime (Kraemer et al., 2015; Magee and Wu, 2017) but it cannot explain all the observed variance in lake stratification or temperature (Kraemer et al., 2015; O'Reilly et al., 2015). Besides morphology, the mixing regime depends on climate and water transparency (Kirillin and Shatwell, 2016) and therefore integrates many of the factors that determine how lakes react to climate change. Transparency influences

the vertical temperature distribution by determining how far solar radiation can penetrate into the water column. Low transparency generally leads to stronger vertical temperature gradients and stratification, a shallower mixed layer depth, and lower deep water and whole-lake temperatures (Hocking and Straskraba, 1999; Persson and Jones, 2008; Thiery et al., 2014a; Yan, 1983). Decreasing transparency (dimming) has been shown to buffer or even potentially reverse climate-induced trends in mean lake temperature (Rose et al., 2016). However, the opposite is true for stratification, where warming

and dimming effects amplify to increase thermal gradients (Kirillin, 2010). The pattern becomes even more complicated with seasonal variations of the effect of transparency on stratification as well as transparency itself (Rinke et al., 2010; Shatwell et al., 2016). A major factor determining water transparency is the trophic level, which can also experience long-term variations due to changing nutrient supply. Thus, the sensitivity of warming to transparency warrants more investigation.

The present study was designed to analyse the effects of seasonality on the response of northern temperate lakes to projected

future warming. We project future changes in thermal characteristics and mixing regime of four temperate lakes with a thermodynamic model forced by an ensemble of climate projections based on a moderate warming scenario, the Radiative Concentration Pathway 4.5 (RCP4.5). To differentiate the potential effect of mixing regime on lake response, we simulate two shallow and two deep lakes, where each pair is similar in terms of morphology but different in terms of transparency and/or mixing regime. The four lakes are located within a distance of ≤ 150 km from each other; hence differences in their

response to climate change are conditioned predominantly by lake properties. The aim of our study is to better understand the internal physical mechanisms determining the response of lakes to a future warmer climate by interaction of vertical mixing, ice formation, and water transparency.

## 2 Methods

### 2.1 Study sites

We consider four lakes located in north-eastern Germany in a temperate continental climate (Fig. 1). Stechlinsee (53.15° N, 13.03° E) and Arendsee (52.89° N, 11.48° E) are deep dimictic groundwater-fed lakes of similar area and depth. Stechlinsee is oligo-mesotrophic and one of the clearest lakes in the region, whereas Arendsee is eutrophic and turbid (Table 1).



Müggelsee (52.44° N, 13.65° E) is a eutrophic, polymictic lake and Heiligensee (52.61° N, 13.22° E) is a hypertrophic, dimictic lake. Both are shallow lakes and located in Berlin.

## 2.2 Lake data

Long term water temperature measurements were available at weekly intervals and 0.5 m depth increments for Müggelsee, in bi-weekly to monthly intervals at 1 m depth increments for Heiligensee, and in weekly to monthly intervals at varying depth increments in both Stechlinsee and Arendsee. In addition, hourly water temperature profiles were available for Müggelsee since 2004. The thermodynamic model was calibrated with long-term temperature measurements from 1979-2014 (Müggelsee), 1979-2001 (Heiligensee), 1991-2014 (Stechlinsee), and 1979-2010 (Arendsee). Stratification duration measurements in Müggelsee were only based on the high frequency temperature measurements from 2004-2014. Secchi depths were measured with a standard 20 cm disk at weekly to monthly intervals at least since 1979 in all lakes. Light extinction in Müggelsee, Heiligensee, and Stechlinsee was calculated using the Lambert-Beer law from simultaneous light measurements at different depths (generally 0.5 m apart) recorded with spherical sensors (Licor, Nebraska). Light extinction in Arendsee was estimated from Secchi depth measurements using the relation of Poole and Atkins (1929) as $\gamma = 2.17 / h_{secchi}$. The constant 2.17 was estimated as the mean of the empirically derived constants of 2.05 (Müggelsee), 2.13 (Heiligensee) and 2.33 (Stechlinsee) (Shatwell et al., 2016).

## 2.3 Modelling

Modelling was performed with the lake temperature and mixing model FLake (Kirillin et al., 2011; Mironov, 2008)—a bulk model of the lake thermal regime specifically designed to parameterize inland waters in climate models and numerical weather prediction systems. FLake is based on a two-layer parametric representation of the vertical temperature structure. The upper layer is treated as well-mixed and vertically homogeneous. The structure of the lower stably-stratified layer, the lake thermocline, is parameterized using a self-similar representation of the temperature profile. The same self-similarity concept is used to describe the temperature structure of the thermally active upper layer of the bottom sediments (Golosov and Kirillin, 2010) and the ice cover (Mironov et al., 2012). The depth of the mixed layer is computed from the prognostic entrainment equation in convective conditions, and from the diagnostic equilibrium boundary-layer depth formulation in conditions of wind mixing against the stabilizing surface buoyancy flux. The integrated approach implemented in FLake allowed combining high computational efficiency with a realistic representation of the major physics behind turbulent and diffusive heat exchange in the stratified water column and similar skill compared to other lake models (e.g., Stepanenko et al., 2014; Stepanenko et al., 2013; Thiery et al., 2014b). As a result, FLake is widely used for representing lakes in land schemes of regional and global climate models, being implemented, for example, into the surface schemes of the Weather Research and Forecasting model (WRF, Mallard et al., 2014) and Consortium for Small-scale Modelling (COSMO; Mironov et al., 2010), HTESSEL (ECMWF; Dutra et al., 2010), SURFEX (Meteo France; Salgado and Le Moigne, 2010), and JULES (UK Met Office; Rooney and Jones, 2010). Thanks to its robustness and computational efficiency, FLake has become the





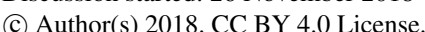

standard choice in climate studies involving the feedbacks between inland waters and the atmosphere, and is used operationally in the NWP models of the German Weather Service (DWD), the European Centre for Medium-Range Weather Forecasts (ECMWF), the UK Met Office, the Swedish Meteorological and Hydrological Institute (SMHI), the Finnish Meteorological Institute (FMI) and others.

FLake is a process-based model with a high level of parameterization, which includes several empirical constants estimated from independent observational and numerical data. The model parameterizations have proven reliable in application to lakes of different morphometry and climatic conditions (Docquier et al., 2016; Kirillin, 2010; Kirillin et al., 2013; Kirillin et al., 2017; Martynov et al., 2010; Shatwell et al., 2016; Thiery et al., 2015; Thiery et al., 2016; Thiery et al., 2014a), so that FLake does not normally require re-tuning for a specific lake. Minor model adjustments were performed based on

observations. The adjustments refer to two model parameters, each of them encompassing multiple effects of lake-specific mechanisms on the thermal stratification.

The first parameter is the rate of change of the self-similar temperature profile shape under the surface mixed layer $\vartheta(\zeta)$. It is assumed to vary with time, so that the integral shape factor $C_{\vartheta} \equiv \int_0^1 \vartheta(\zeta)\mathrm{d}\zeta$ fluctuates between the two asymptotic bounds $C_{\vartheta}^{max}$ and $C_{\vartheta}^{min}$ with a characteristic time scale $t^*$ as

$$\frac{\mathrm{d}C_{\vartheta}}{\mathrm{d}t} \propto \frac{C_{\vartheta}^{max} - C_{\vartheta}^{min}}{t_*}, \qquad\qquad (1)$$

The parameterization incorporates mixing in the lower stratified layer (hypolimnion) by intermittent shear turbulence and breaking internal waves. The time scale $t^*$ is parameterized as a function of the thickness of the hypolimnion, the strength of stratification across it, and the rate of mixing at the top of the thermocline (see Mironov (2008) for details). Observational data on temporal variations of $C_{\vartheta}$ in several lakes demonstrated that $t_*$ tends to vary among them depending on lake depth

(Kirillin unpubl.). This parameter was increased from 0.04 to 0.22 for the shallow Müggelsee, aimed to mimic the additional vertical mixing produced by the flow of the River Spree through the lake, which is not explicitly accounted for by 1-dimensional modelling. The second parameter is the rate of change of the wind-mixed layer depth, which is assumed in the model to relax exponentially to the equilibrium depth determined following Zilitinkevich and Mironov (1996). The parameter is suggested to have the order of magnitude $10^{-2}$ (Mironov 2008) and was set to 0.025 for all lakes based on

observed mixed layer dynamics.

## 2.4 Climate warming scenarios

FLake was forced with an ensemble of continuous 21[st] century climate projections from the Coordinated Regional climate Downscaling Experiment for Europe (EURO-CORDEX; Kotlarski et al., 2014). The climate projections were based on an intermediate greenhouse gas concentration trajectory (RCP4.5), representing a moderate climate warming scenario with an

end-of-century, top-of-the-atmosphere radiative forcing of 4.5 W m$^{-2}$ compared to the pre-industrial period (IPCC 2014).



The ensemble was assembled from different downscaled global climate models (MPI-ESM-LR, EC-EARTH, CanESM2, CNRM-CM5, CSIRO-Mk3-6-0, GFDL-ESM2M, MIROC5, NorESM1-M) each providing lateral boundary conditions to the RCA4 regional climate model. In addition, the MPI-ESM-LR global climate model (GCM) was downscaled by the CCLM4-8-17 and REMO2009 regional climate models, and the EC-EARTH GCM was downscaled by HIRHAM5 and RACMO22E, yielding an ensemble of 12 climate projections (2006 – 2100) for each lake. All simulations were performed at a horizontal resolution of 0.44°. For the analysis we selected the model pixel containing the lake under consideration.

## 2.5 Temporal downscaling of meteorology

The FLake forcing variables included solar radiation ($I_R$), near-surface air temperature ($T_a$), 10 m wind speed ($U_{10}$), humidity ($e_a$) and cloud fraction ($N$). These variables were available at daily resolution in the climate projections, and were downscaled to sub-daily resolution with the same daily mean to account for diurnal forcing. Solar radiation and wind speed were downscaled assuming a sine course during the day, with a constant minimum value during the night, according to the model:

$$p = p_{\min} + p_{\max} \sin\left(\frac{\pi(t - lag - t_{rise})}{t_{set} - t_{rise}}\right)^m , p \geq p_{min} \geq 0 \qquad (1)$$

where $p$ is the solar radiation or wind speed as a function of time of day ($t$, in hours), $p_{\min}$ and $p_{\max}$ are the night-time minimum and daytime maximum values, respectively, $t_{rise}$ and $t_{set}$ are the times of sunrise and sunset, respectively, $lag$ is the time lag of the maximum value behind solar noon (half way between $t_{rise}$ and $t_{set}$), and $m$ is a shape constant. The constant $m$ was empirically determined to be 1.3 based on high resolution weather data from the Potsdam weather station near Berlin. Sunrise and sunset times were determined at the coordinates of each study lake on each day of the year with a NOAA-algorithm using the R package maptools (Bivand and Lewin-Koh, 2016). The daily mean is given by

$$\overline{p} = p_{\min} + (p_{\max} - p_{\min}) \frac{(t_{set} - t_{rise})}{24} \Gamma\left(\frac{m+1}{2}\right) \left(\sqrt{\pi} \Gamma\left(\frac{m}{2} + 1\right)\right)^{-1} \qquad (2)$$

For solar radiation, $lag = 0$ and $p_{\min} = 0$. For wind speed, $lag = 1$ h based on empirical observations and $p_{\min}$ is non-zero. Accordingly, the daily course of solar radiation $p$ for a prescribed daily mean $\bar{p}$ from the climate projections is given by calculating $p_{\max}$ with Eq. (2) and substituting it into Eq. (1). For wind speed, the daily course is defined from the daily mean given the daily amplitude $p_{\max} - p_{\min}$, which was empirically estimated from the Potsdam weather data and varied seasonally. Sub-daily air temperatures were interpolated between a daytime maximum assumed at 2 pm and night-time minimum assumed at 2 am using a cubic spline, where the daily temperature amplitude $\Delta T_a$ (K) was estimated from the daily mean temperature $\overline{T_a}$ (°C) using the empirical relationship $\Delta T_a = 0.345 \overline{T_a} + 5.5$ based on the Potsdam data. Sub-daily humidity and cloud fraction were linearly interpolated between the daily mean values.



### 2.6 Extinction scenarios

We performed lake model simulations with different scenarios of seasonal extinction, which in these lakes is primarily determined by phytoplankton chlorophyll (Shatwell et al., 2016). Here the thermodynamic model was modified to account for seasonally variable extinction, following a typical bimodal pattern according to the Plankton Ecology Group (PEG)

model (Sommer et al., 2012; Sommer et al., 1986), as described in detail in Shatwell et al. (2016). A base seasonal extinction pattern was derived from long term observations of extinction and/or Secchi depth in each study lake, preserving important seasonal characteristics including long-term annual mean extinction (see Table 1), minimum extinction, and timing and magnitude of the spring and summer blooms (Fig. 2). The spring and summer blooms were described by Weibull and Gauss curves, respectively.

Using these base extinction patterns as the control scenario, we simulated the hydrodynamics of the four lakes for the period 2006-2100 forced by the climate projection ensemble. To test the sensitivity of lake warming response to water clarity, we then repeated these simulations with a modified mean extinction coefficient. The mean extinction coefficient was varied by scaling the whole seasonal extinction pattern, affecting mean, minimum and peak extinction equally. A principal component analysis on long-term extinction coefficient measurements in Müggelsee and Heiligensee showed that this method best

represented the natural variation in the seasonality of extinction (Shatwell et al., 2016).

### 2.7 Model setup and calibration

FLake was configured for each lake by setting mean lake depth and fetch, the temperature at the bottom of the thermally active sediment layer (set to the historical long term mean water temperature at the lake bottom), and the extinction coefficient as described above. The model was manually calibrated for each lake by adjusting $t_*$ with a constant.

Additionally, in the case of shallow Müggelsee, the rate of change of the mixed layer depth due to wind forcing was changed to account for the additional vertical mixing produced by the riverine throughflow (see Sect. 2.3 for details). Hindcast simulations were forced with meteorology from ERA-Interim gridded reanalyses produced by the European Centre for Medium-Range Weather Forecasts (Dee et al., 2011). Using this meteorological forcing as a basis for the model calibration was preferred over using (i) in-situ meteorology, as the future projections are obtained by forcing FLake with a gridded

simulation product rather than in-situ information, and (ii) the EURO-CORDEX reanalysis downscaling simulations, as calibrated model parameters would in that case differ between model projections. For consistency the ERA-Interim driven simulations were downscaled from daily means to sub-daily values using the same procedure as for the climate projections described above.

Model performance was assessed by comparing the modelled and observed surface temperature ($T_s$), bottom temperature

($T_b$), and, for the seasonally stratified lakes (Heiligensee, Stechlinsee and Arendsee), the mean summer mixed layer depth ($h_{mix}$), and timing and duration of seasonal stratification. As different measures of goodness of fit, we examined model bias (Eq. 3), centred root mean square error ($RMSE_c$, Eq. 4), normalized root mean square error ($RMSE_n$, Eq. 6), as well as





normalized standard deviation ($\sigma_{norm}$, Eq. 5), where $m_i$ and $o_i$ are the modelled and observed values, respectively, and $\bar{m}$ and $\bar{o}$ are their means:

$$bias = \bar{m} - \bar{o} \tag{3}$$

$$RMSE_c = \sqrt{\frac{1}{n} \sum_{i=1}^{n} ((m_i - \bar{m}) - (o_i - \bar{o}))^2} \tag{4}$$

$$RMSE_n = \sqrt{\frac{1}{n} \sum_{i=1}^{n} (o_i - m_i)^2} \Big/ \sqrt{\frac{1}{n} \sum_{i=1}^{n} o_i^2} \tag{5}$$

$$\sigma_{norm} = \sqrt{\sum_{i=1}^{n} (m_i - \bar{m})^2 \Big/ \sum_{i=1}^{n} (o_i - \bar{o})^2} \tag{6}$$

**2.8 Data handling and statistics**

$T_s$ was calculated from field data as the mean temperature in the $0 - 1$ m layer. $T_b$ was calculated as the mean temperature at the mean depth of each lake. Stratification was inferred when the absolute difference between $T_s$ and $T_b$ exceeded 0.5 °C. The lake was assumed to be mixed when it was not stratified. The duration of seasonal stratification was defined as the length of longest uninterrupted stratification period each season. Similarly, the duration of overturn was defined as the length of the longest uninterrupted period of mixing in each season following the main stratification events. Winter was defined as January to March, spring as April to June, summer as July to September, and autumn as October to December. Trends were assessed using linear regression over time. Differences between transparency treatments were assessed by comparing decadal means of a particular variable for each climate projection in the ensemble, yielding 12 means per decade over 9 decades from 2010 to 2100. Comparisons were made using ANCOVA with decade as the covariate. Normality of residuals was assessed with quantile-quantile plots, homogeneity of variance with plots of residuals versus fitted values, and outliers with plots of Cook's distances. Increasing variance with the mean was resolved by log-transformation. One day was added to ice-cover duration to avoid zero values before log-transformation. Outliers with a Cook's distance > 1 were excluded from statistical analyses. Small deviations from normality were tolerated. All statistical analyses were performed with R version 3.3.0 (R Core Team, 2016).

**3 Results**

**3.1 Model validation**

The thermodynamic model forced by the ERA-Interim reanalysis data performed well at reproducing the observed thermodynamic characteristics of the study lakes (Fig. 3). The model predicted water temperature with good precision as evident in $RMSE_c$ values generally less than 1.5 °C for surface and bottom temperatures (Table 2). The model also reproduced observed temperature variability well with $\sigma_n$ close to one, except in bottom temperatures of the two deep lakes Stechlinsee and Arendsee, which are quite invariant with a standard deviation of 0.86 and 0.99 °C respectively. However the





model had a systematic tendency to underestimate $T_s$ in all lakes by 0.7 – 1.8 °C. The model also adequately reproduced the stratification characteristics of the three seasonally stratified lakes with a centred error of 1 – 2 m for $h_{mix}$ and about 2 – 3 weeks for stratification duration, noting that the sampling interval (and thus measurement error) for stratification duration was generally 4 weeks. The model systematically underestimated the stratification duration by about one month, probably

because the surface temperatures were also underestimated.

### 3.2 Meteorology trends

Air temperatures in the RCP4.5 forcing ensemble increased on average by 0.16°C per decade over all lakes. These increases were strongest in winter and spring (0.23 – 0.30 °C decade$^{-1}$) and weakest in summer (0.04 °C decade$^{-1}$, Fig. 4). Mean annual radiation decreased in the ensembles by 0.29 W m$^{-2}$ decade$^{-1}$. The overall decrease was unevenly distributed seasonally, with

a strong decrease projected for summer (1.65 – 1.70 W m$^{-2}$ decade$^{-1}$) but an increase projected for winter (1.1 W m$^{-2}$ decade$^{-1}$). Annual mean water vapour pressure was projected to increase by 0.1 Pa decade$^{-1}$ with stronger increases in winter and spring than in summer and autumn. The projected changes in wind speed were negligible.

### 3.3 Lake response to warming

In model runs with the historical, baseline extinction, these projected climate trends resulted in an increase in annual surface

water temperatures (higher rates of increase in deeper lakes) in all lakes for the simulation period 2006 – 2100 (Fig. 5, Table 3). Bottom temperatures increased in all lakes, with faster warming rates projected in the two shallower lakes Müggelsee and Heiligensee. Accordingly, mean lake temperatures increased in all lakes at 0.10 – 0.11 °C decade$^{-1}$.

$T_s$ increased most rapidly during the winter in the two shallower lakes and during spring in the two deeper lakes. On the other hand, mean summer $T_s$ increased only marginally in Stechlinsee (0.02 °C decade$^{-1}$) and not at all in the other lakes

(Fig. 6). $T_b$ increased most rapidly in winter. Moreover, the higher $T_b$ in April and May, when stratification typically began, persisted until the end of stratification (September in Heiligensee, and November-December in deeper Stechlinsee and Arendsee, Fig. 6). Accordingly, deep water warmed faster than surface water during summer (and autumn in the deep lakes), which led to a lower vertical temperature gradient in summer, and consequently weaker stratification. Furthermore, annual maximum surface temperature increased and occurred earlier in summer (Fig. 7, Table 3).

Winter and spring warming caused stratification to begin earlier in all lakes in the model simulations (Fig. 7). This was also reflected in a shallower mean $h_{mix}$ during spring in the deep lakes Stechlinsee and Arendsee. The end of stratification also advanced slightly due to the slightly smaller vertical temperature gradients in summer, but not as rapidly as stratification onset, so that overall stratification duration increased. Mean summer $h_{mix}$ increased in 3 lakes by less than 0.05 m decade$^{-1}$, which is practically negligible. Winter stratification and ice cover were more intermittent in the two shallow lakes than in the

two deeper lakes. Winter and spring warming mainly caused winter stratification and ice cover to end earlier, reducing the overall duration (Fig. 7). Interestingly, most of the changes in winter stratification and ice cover were projected to occur in the first half of the 21st century, with little change in the second half.



The projected increase in stratification duration was insufficient to cause Müggelsee to stratify continuously from spring to the end of summer (median duration in the decade 2090-2099: 102 days) and thus shift from a polymictic to a dimictic regime. On the other hand, the deeper dimictic lakes Stechlinsee and Arendsee were projected to be at the transition to a monomictic regime by the middle of the 21$^{st}$ century. In the decade 2050-2059, the model projected that Stechlinsee and Arendsee would have no winter stratification in 33% and 55% of winters and be ice-free in 76% and 93% of winters, respectively. By comparison, in the decade 2010-2019, Stechlinsee and Arendsee were projected to have no winter stratification in 13% and 24% of winters, and be ice-free in 58% and 75% of winters, respectively. The mean winter bottom temperature in the decade 2090-2099 was projected to be $3.96 \pm 0.93$ °C in Stechlinsee (compared to $3.03 \pm 0.79$ °C in 2010-2019), and $4.48 \pm 0.93$ °C in Arendsee (compared to $3.62 \pm 0.73$ °C in 2010-2019).

Annual mean whole-lake water temperature correlated with annual mean air temperature. At any given annual mean air temperature, Müggelsee was warmest, followed by Heiligensee, Stechlinsee and Arendsee. Moreover, at a given annual mean air temperature, all lakes were colder following winters with ice cover than without ice cover, where the temperature differences increased with mean lake depth.

## 3.4 Interaction between warming and transparency

To investigate the sensitivity of different lake types to a change in transparency combined with warming, we modelled the projected warming effects (2006 – 2100) in the given lakes with mean extinction ranging from extremely clear ($\gamma = 0.05$ m$^{-1}$) to extremely turbid ($\gamma = 3$ m$^{-1}$). The effect of transparency and its interactions with warming were highly non-linear and the responses were very sensitive to lake type. There were transparency ranges where transparency had a very large effect on thermal structure, and ranges where it had little effect at all. In general, changes in transparency had a strong effect on stratification duration and vertical temperature structure at intermediate extinction (ca. $0.4 – 1.1$ m$^{-1}$) in the moderately shallow lakes Müggelsee and Heiligensee (Fig. 8a, b), and at very low extinction (< 0.3 m$^{-1}$) in the deep lakes Stechlinsee and Arendsee (Fig. 8c, d). Outside these ranges, changes in extinction had a much smaller effect: at extinctions above about 1.1 m$^{-1}$ in the two shallower lakes (or > 0.3 m$^{-1}$ in Stechlinsee) a further increase in extinction no longer increased stratification duration, but rather decreased it gradually due to a very gradual increase in summer bottom temperatures. Accordingly, at extinction values higher than the historical means in the four study lakes (vertical dashed lines in Fig. 8, see also Table 3), the trends in stratification duration due to warming were stable and relatively insensitive to changes in transparency. Thus, an increase in extinction combined with warming is unlikely to push Müggelsee from a polymictic to a dimictic regime. However, the simulations also suggested that clearer shallow lakes, which are close to the transition between polymictic and dimictic regimes, can potentially switch mixing regimes because interactions between warming and dimming are stronger. Altogether, the model suggested that changes in stratification duration due to changes in transparency can be expected at optically normalised depths $\gamma \times h_{\mathrm{mean}} <$ 5 to 8 (Fig. 9).

The model predicted that an increase in extinction in the four study lakes above their historical values would not influence whole-lake warming rates ($T_{\mathrm{m}}$) induced by climate change (Fig. 8i- l). However, the model did suggest that there would be



an interaction between warming and dimming at extinctions considerably lower than the historical extinction values observed in these lakes. In the deep lakes, dimming was predicted to decrease $T_m$ at extinctions below about 0.3 m$^{-1}$ (Fig. 8c, d). According to simulations, increasing extinction from 0.1 to 0.3 m$^{-1}$ would approximately cancel the expected effect on $T_m$ of 90 years of climate warming. The shallower lakes behaved differently to the deep lakes at low extinction. In Müggelsee, $T_m$ increased with increasing extinction up to about 0.5 m$^{-1}$ (Fig. 8a). In Heiligensee the model suggested a strong interaction between warming and dimming up to an extinction of about 0.5 m$^{-1}$, where dimming would decrease $T_m$ in the warmer climate at the end of the 21$^{st}$ century but increase $T_m$ in the cooler climate of today (Fig. 8b).

Whereas the model predicted that warming would have a small (deepening) effect on mixed layer depth, it predicted that an increase in extinction would have a stronger (shoaling) effect (Fig. 8e – h). Again, a change in extinction would have a much stronger effect at intermediate extinctions in the shallower lakes and a low extinction in the deep lakes.

## 4 Discussion

### 4.1 Warming trends

Our projections suggest that surface waters in all modelled lakes will warm at about 75-90% of the rate of air temperature increase, with surface temperatures in the deeper lakes increasing faster than the shallower lakes. This is similar to other modelling studies based on climate projections, which found warming rates of 70-85% of air temperature trends (Butcher et al., 2015; Schmid et al., 2014). The relatively uniform results obtained from climate and lake modelling studies are not entirely consistent with empirical evidence, which shows that warming rates of surface water relative to air can be more variable (O'Reilly et al., 2015; Torbick et al., 2016).

### 4.2 Influence of seasonality and stratification

We found that seasonality of warming played an important role and we concur with Winslow et al. (2017) that summer lake warming rates are not representative of warming in other seasons. In fact, our projections suggest that the study lakes will warm predominantly in winter and spring, and only marginally in summer and autumn. These seasonal patterns largely reflect the seasonality of air temperature trends in the warming ensemble. The leftward seasonal skew of surface water temperature increase towards spring and our estimates that the summer maximum temperature should occur increasingly earlier in the year imply that the number of days per year when the surface temperature exceeds a certain threshold will increase. Accordingly, although summer warming was slow, the length of the bathing season will likely increase and along with it both the recreational value and the recreational pressure on lakes.

Stratification and ice cover restrict vertical heat transport and thus influence heat storage in lakes. Therefore they should modulate how lakes respond to warming. This was evident in our simulations because, at the same annual mean air temperature, lakes were colder in years with ice cover (dimictic regime) than in years without ice (monomictic regime). This can be explained in part by the thermal inertia from the previous season (Crossman et al., 2016), and in part due to altered





heat accumulation in deep water during mixing preceding summer stratification. The latter was demonstrated by the relatively steep trends in bottom temperature during winter simulated in the four study lakes. These effects were also observed empirically in Stechlinsee, which was thermally polluted by a nuclear power plant, preventing ice cover and shifting the mixing regime from dimictic to monomictic (Kirillin et al., 2013). Here the waste heat input during winter

mixing was stored in deep water until the following winter and increased the mean lake temperature, whereas the waste heat input during stratification was confined to the surface layer and quickly lost to the atmosphere. The effect of summer stratification on heat transport and storage explains why the simulated mean lake temperature increased faster in the polymictic lake Müggelsee than in seasonally stratified Heiligensee, although these lakes have very similar mean depths. We conclude that not just morphology, but also mixing regime modulates how lakes respond to climate warming. Moreover, we

conclude that seasonal differences in climate warming rates combined with the strongly non-linear sensitivity of lakes to monthly temperature changes determine how lakes overall react to warming.

## 4.3 Mixing regime shifts

The comparatively rapid increase in winter air temperature was projected to strongly affect ice cover and winter stratification, and push the two deeper dimictic lakes (Stechlinsee and Arendsee) towards a monomictic regime. Such shifts

have been projected as a consequence of climate warming for temperate stratified lakes (Ficker et al., 2017; Kirillin, 2010; Livingstone, 2008). Both of these lakes switched regularly between monomixis and dimixis and the climate-induced transition between regimes is projected to be gradual. Particularly the slightly deeper Arendsee will be rarely ice covered in the coming decades and will experience a predominantly monomictic regime and mean winter bottom temperatures generally above 4°C. Our finding that almost all of the change in winter stratification and ice cover will occur in the first half

of the 21[st] century is in part because the winter air temperature in the ensemble is projected to increase twice as fast then as in the second half of the 21[st] century, and also because of the highly non-linear response of ice cover to air temperature changes. Our model projections suggested that lake depth was the most important factor regulating ice cover duration, as expected (Kirillin et al., 2012). Moreover, the timing of ice off was more sensitive to warming than ice on, which concurs with several studies (Benson et al., 2012; Crossman et al., 2016; Vavrus et al., 1996). The sensitivity analysis suggested that

transparency would not influence ice cover in the modelled lakes, but may have an effect in extremely clear lakes. Other modelling studies suggested that transparency can affect the timing of ice-on (Heiskanen et al., 2015) but may have little effect on ice-off timing (Fang and Stefan, 2009). The disappearance of ice will increase evaporation rates, as well as the exchange of heat and gases with the atmosphere.

According to our simulations, summer stratification duration in Müggelsee is likely to increase, but not enough to switch the

lake from a polymictic to a dimictic regime within the current century. An increase in the frequency of stratification events is likely to increase the occurrence of nuisance cyanobacterial blooms and influence nutrient cycles, as was already demonstrated in this lake (Wagner and Adrian, 2009). The warming effect on stratification duration and mixed layer depth was relatively small in comparison with the potential effect of a change in transparency (Fig. 8a, e). Thus, lakes at the



transition between polymictic and dimictic regimes may shift more due to changes in transparency or water level than due to warming alone.

## 4.4 Mixed layer depth

An important question that has not been conclusively answered is how climate change will affect the mixed layer depth. Our
simulations suggested that climate warming would increase the mixed layer depth in stratified lakes, but the projected increase of less than 0.05 m decade$^{-1}$ should be considered negligible. Other lake model simulations have also predicted the mixed layer to deepen with warming (Missaghi et al., 2017). In contrast, ocean models have consistently projected a shoaling of the mixed layer (Capotondi et al., 2012; Jang et al., 2011; Sen Gupta et al., 2009) but empirical evidence does not confirm this (Somavilla et al., 2017). In lakes, empirical data show that the mixed layer can deepen (Flaim et al., 2016; Kraemer et
al., 2015), remain relatively constant (Ficker et al., 2017; Kirillin et al., 2013) or possibly shoal (Saros et al., 2016) as a result of warming. In our simulations, the mixed layer depth was much more sensitive to transparency than warming, which concurs with empirical data (Flaim et al., 2016; Saros et al., 2016). We conclude that there is little physical evidence that climate warming causes the mixed layer to become shallower and suggest that observed changes in mixed layer depth may be more due to changes in transparency or wind speed.

## 4.5 Transparency – warming interactions

A major finding of our study was that interactions between global warming and changes in transparency are highly non-linear. Thus, we confirm our initial hypothesis that warming and dimming act in synergy to increase stratification duration but oppose each other to stabilize mean lake temperature. However, we add that this only applies within certain extinction ranges which vary with lake mixing regime. In general, interactions between a change in transparency and warming occurred
at very low to moderate extinction. The reason for the non-linearity is probably the extent to which transparency influences the mixed layer depth and the extent to which the mixed layer interacts with the lake bottom. Transparency is the major determinant of mixed layer depth in small, wind sheltered lakes but its effect diminishes with increasing lake size and wind speed (Kirillin and Shatwell, 2016).

According to the sensitivity analysis, a change in transparency was projected to affect mean lake temperature most strongly
at very high transprency, where the nature of the effect depended on the lake mixing regime. At high transparency (low extinction coefficient), dimming increased lake temperature in polymictic Müggelsee because increased absorption of radiation affected the whole water column (Fig. 8). On the other hand, dimming decreased lake temperature in the stratified lakes Arendsee and Stechlinsee because less radiation reached the deep waters. Heilgensee was the most sensitive because dimming was projected to alter the mixing regime from polymictic to stratified. Although the model suggested that a change
in transparency would have strong effects on lake thermal properties, the four lakes we simulated were predicted to be too turbid in their present state to be substantially influenced by a further decrease in transparency. This may be because shallower lakes are naturally more turbid than deep lakes, so that lakes in which a substantial amount of incoming radiation





is able to reach large areas of the lake bottom are rare. Very shallow lakes where radiation can reach most of the lake bottom tend to be dominated by macrophytes, which themselves strongly alter stratification and mixing conditions (Vilas et al., 2017). Transparency is affected by anthropogenic activities (Jeppesen et al., 2005) and also climate change (Larsen et al., 2011), so investigations into how lakes respond to global warming should consider these complex interactions.

## 4.6 Model reliability

The model errors in this study were relatively small compared to other studies (for discussion and comparison see Shatwell et al. (2016)), providing confidence in the results. There were however some systematic errors, which we attempted to alleviate by focusing on comparative analyses, such as trends. Our finding that mean bottom water temperatures in summer should increase under higher extinction in the stratified lakes was unexpected according to empirical evidence (Snucins and Gunn, 2000). The reason for this result in the simulations was apparently diurnal heating and cooling cycles, which increased the downward transport of heat into the hypolimnion during summer because this effect was present when the model time step was decreased from 6 to 3 hours, but it disappeared when the time step was increased to 1 day (results not shown). A comparison with empirical data suggests that this effect was probably overestimated by the model because a decrease in transparency with a weaker spring clear water phase caused an increase in summer surface-bottom temperature differences in lakes Müggelsee and Heiligensee (Shatwell et al., 2016).

## 5 Conclusion

Altogether our study suggests that warming over the next century will gradually shift many temperate dimictic lakes towards a predominantly monomictic regime, particularly since projected warming rates were highest in winter and spring, with deeper lakes shifting before shallower ones. On the other hand, shifts from polymixis to dimixis are more likely to occur due to a change in transparency or depth than due to climate warming alone. Dimming trends can buffer lake warming trends but amplify increases in stratification duration. However, this effect will probably be limited to relatively clear lakes, particularly those at the transition between polymictic and stratified regimes. Lake mixing regime is an important modulator of climate change impacts on lakes and it is necessary to understand how the seasonality of warming interacts with stratification and ice cover to interpret differences in lake warming trends.

## Code and data availability

The source code of the model FLake is freely (MIT license) available online at www.lakemodel.com. The observational data are stored in the database IGBFRED (https://www.igb-berlin.de/en/freshwater-research-and-environmental-database) subject to the data policy of the Leibniz-Institute of Freshwater Ecology and Inland Fisheries (IGB). Regional climate projections



from the EURO-CORDEX are publicly available through the Earth System Grid Federation (ESGF; https://esgf-data.dkrz.de/). The raw output of the lake modeling is available from the corresponding author by request.

**Author contribution**

GK conceived the study and modified the model with TS. WT provided the climate projections. TS performed the simulations and analysed the results. All authors contributed to interpreting the results. TS wrote the manuscript with contributions from GK and WT.

**Competing interests**

The authors declare that they have no conflict of interest.

**Acknowledgements**

We are grateful to the World Climate Research Programme (WRCP) for initiating and coordinating the CORDEX-Africa initiative, to the modelling centres for making their downscaling results publicly available through ESGF, and to the European Centre for Medium-Range Weather Forecasts (ECMWF) for providing access to ERA-Interim. We thank the Leibniz-Institute of Freshwater Ecology and Inland Fisheries for providing long-term lake data. We also thank the German Meteorological Service (DWD) for providing high resolution weather data and some temperature data for Stechlinsee. TS and GK were funded by a German Research Foundation grant (DFG No. KI-853/7-1). GK was additionally supported by the DFG project IceBound (KI-853/11-1). WT was supported by an ETH Zurich postdoctoral fellowship (Fel-45 15-1).

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

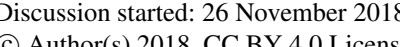

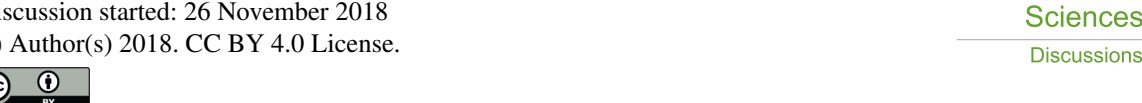

**Figures**

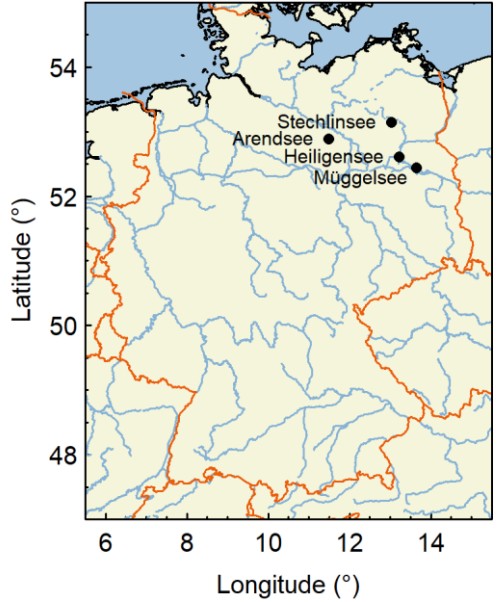

Fig. 1: Locations of the study lakes in Germany.


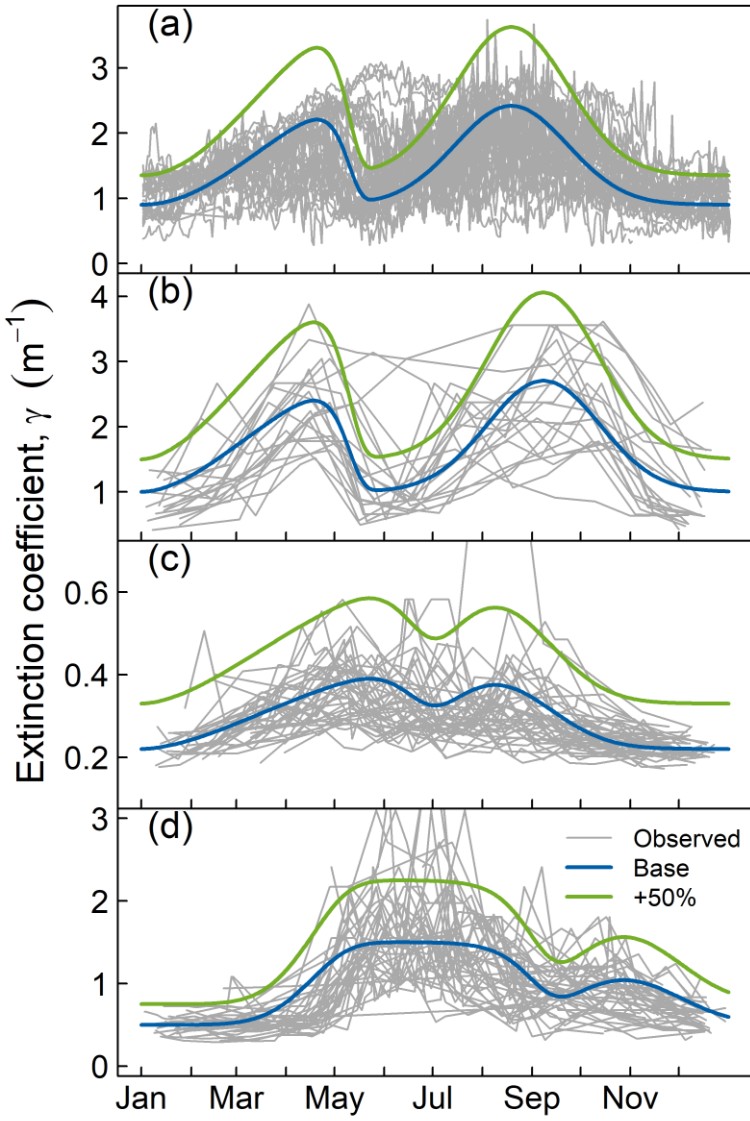

Fig. 2. Seasonal course of extinction (γ) in Müggelsee (a), Heiligensee (b), Stechlinsee (c) and Arendsee (d). Grey lines show superimposed observed extinction during individual years, thick blue lines show the idealized seasonal course of extinction used in model simulations with Flake (base scenario). Green lines show base extinction increased by 50%. Panels a and b modified from Shatwell et al. (2016).





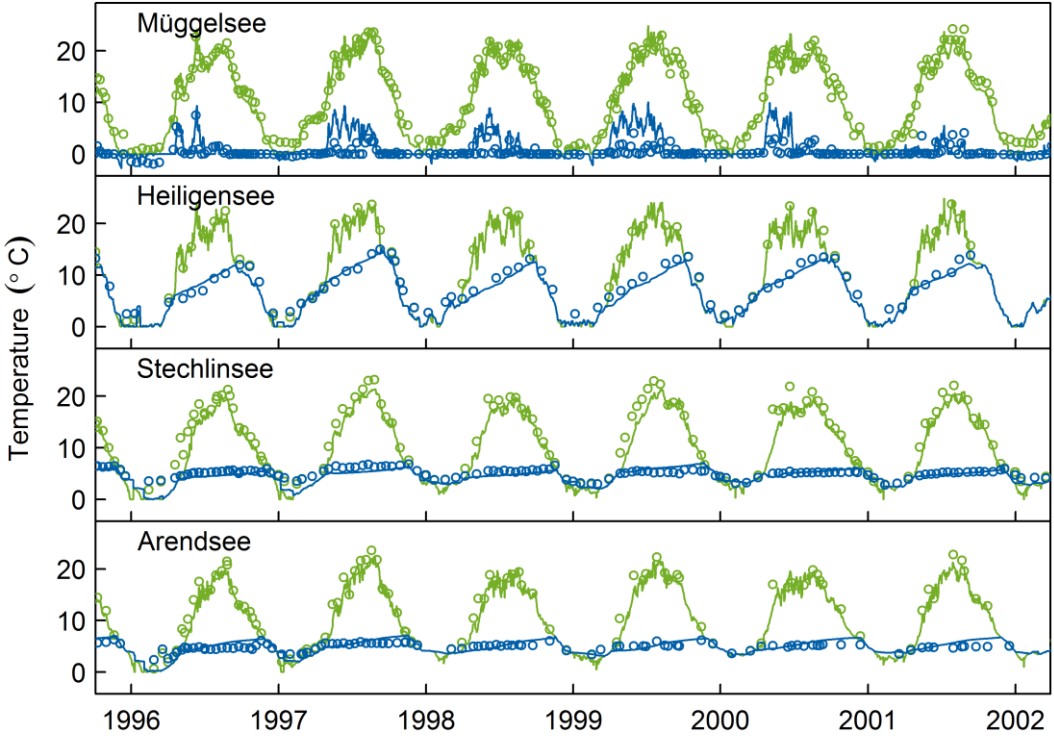

**Fig. 3: Comparison of modelled (lines) and observed temperatures (points) in Müggelsee (a), Heiligensee (b), Stechlinsee (c), and Arendsee (d). Green lines and symbols show surface temperatures, blue lines and symbols show bottom temperatures (b – d) or surface-bottom temperature differences (a).**

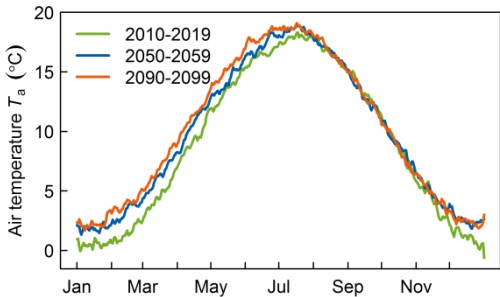

**Fig. 4: Projected changes in over-lake near-surface air temperature under RCP4.5. Lines show decadal means over all climate projections and all four locations.**





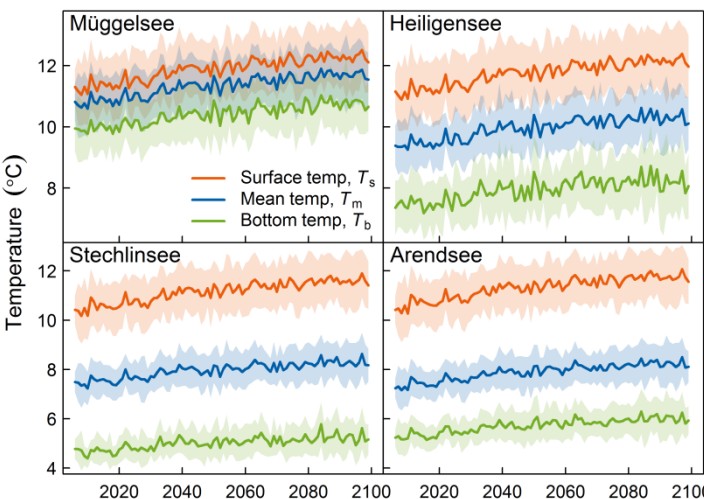

**Fig. 5: Projected evolution of surface, mean and bottom temperatures in the four study lakes. Shading indicates 1 standard deviation either side of the mean.**





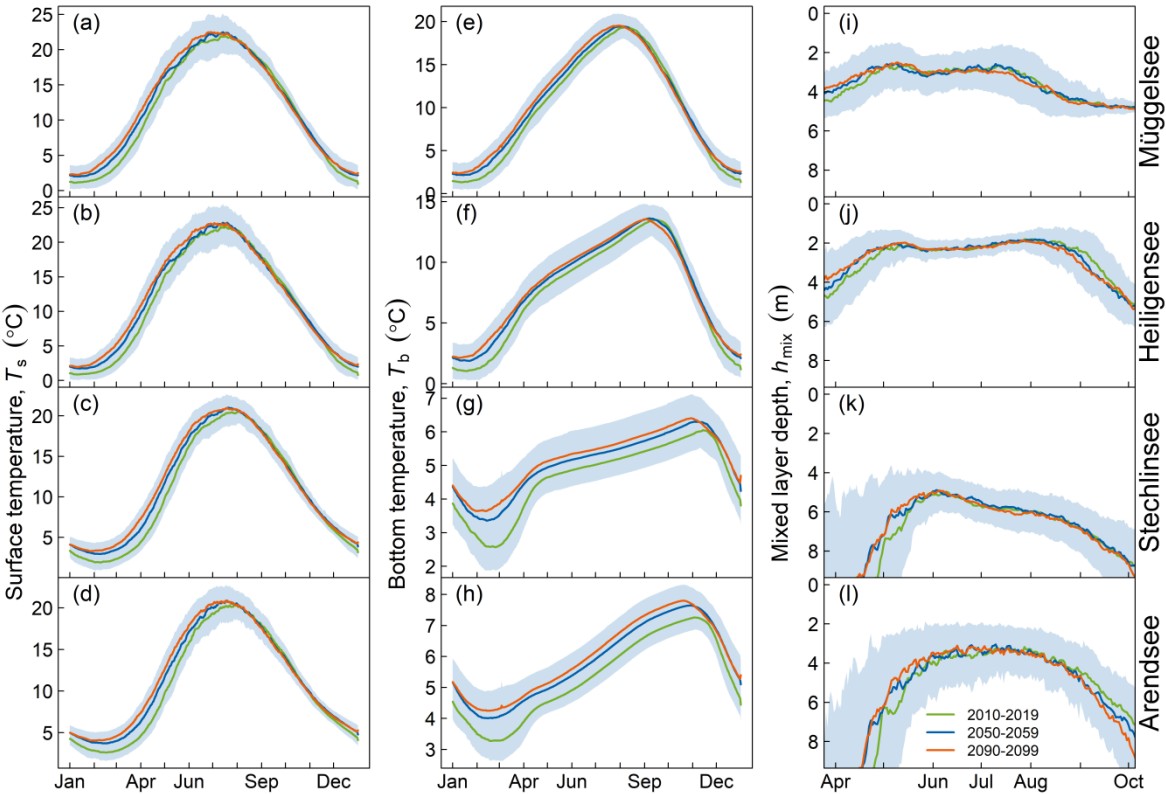

**Fig. 6: Modelled seasonal cycle of surface (a – d) and bottom temperature (e – h) and mixed layer depth (i – l) in Müggelsee (a, e, i), Heiligensee (b, f, j), Stechlinsee (c, g, k) and Arendsee (d, h, l). Lines indicate decadal means and shading denotes 1 standard deviation each side of the mean.**



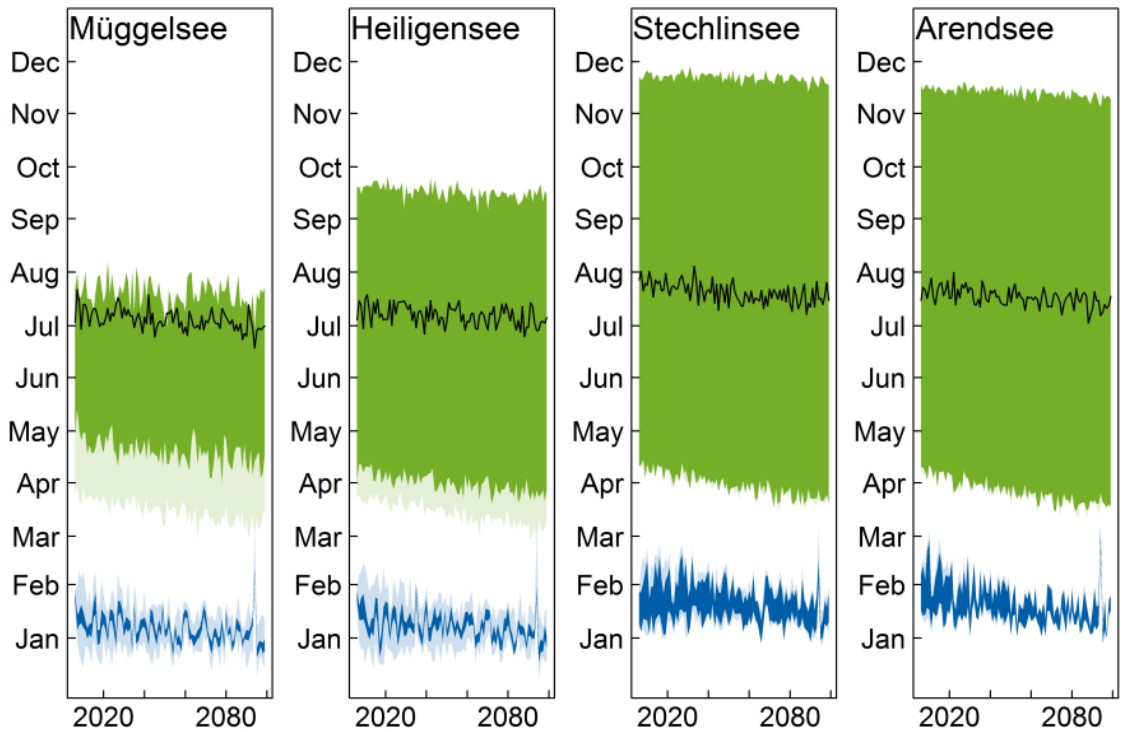

**Fig. 7: Projected trends in stratification in the four study lakes. Green shading shows summer stratification, blue shows winter stratification. Lighter shading indicates intermittent stratification. The black line shows the timing of maximum surface temperature. The model was forced with the baseline (historical) transparency.**





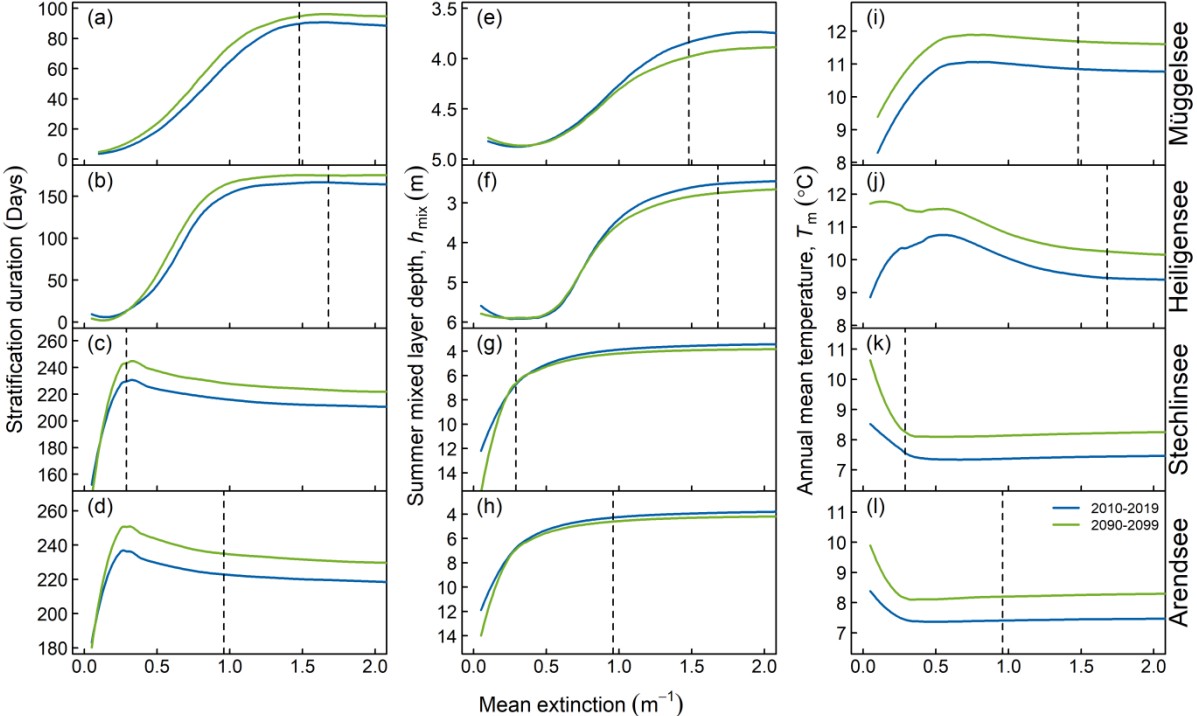

**Fig. 8: Interacting effects of extinction and warming on stratification duration (a – d), mean mixed layer depth in summer (e – h), and annual mean temperature (i – l) in Mügglesee (a, e, i), Heiligensee (b, f, j), Stechlinsee (c, g, k), and Arendsee (d, h, l). Lines have been loess-smoothed and show ensemble means for the decades 2010-2019 (blue) and 2090-2099 (green) as a function of extinction. Black vertical dashed lines show the long-term measured extinction in each lake.**



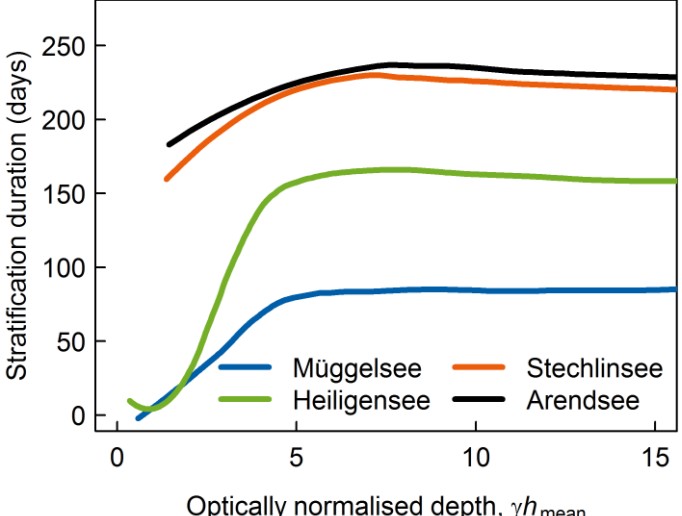

**Fig. 9: Stratification duration vs optically normalised depth (mean extinction ($\gamma$, m$^{-1}$) × mean depth ($h_{mean}$, m), dimensionless) in lakes Müggelsee, Heiligensee, Stechlinsee and Arendsee. The curves are loess-smoothed and are based on ensemble simulations with the baseline extinction in each lake for the 2050 – 2059 period.**





**Tables**

**Table 1: Characteristics of the four study lakes.**

|  | Müggelsee | Heiligensee | Stechlinsee | Arendsee |
|---|---|---|---|---|
| Mean/Max depth (m) | 4.9 / 8.0 | 5.9 / 9.5 | 23 / 69 | 29 / 51 |
| Area (km$^2$) | 7.3 | 0.3 | 4.3 | 5.1 |
| Typical fetch (m) | 4000 | 1000 | 2000 | 4000 |
| Secchi depth (m) | 2.0 ± 0.3 | 1.8 ± 0.4 | 8.6 ± 0.7 | 3.0 ± 1.4 |
| Mean extinction (m$^{-1}$) | 1.48 ± 0.31 | 1.68 ± 0.48 | 0.29 ± 0.03 | 0.96 ± 0.55 |
| Mean stratification duration (d) | 16 ± 7.4 | 169 ± 17 | 250 ± 16 | 232 ± 14 |
| Mixing regime | Polymictic | Dimictic | Dimictic | Di/monomictic |
| Trophic state | Eutrophic | Hypertrophic | Oligotrophic | Eutrophic |





**Table 2: Goodness of fit statistics of the four study lakes. Stratification statistics not shown for polymictic Müggelsee.**

| | Lake | Bias | $RMSE_c$ | $RMSE_n$ | $\sigma_n$ | $n$ |
|---|---|---|---|---|---|---|
| $T_s$ | Müggelsee | -0.67 | 1.02 | 0.07 | 1.13 | 1335 |
| | Heiligensee | -1.07 | 1.04 | 0.09 | 1.18 | 383 |
| | Stechlinsee | -1.83 | 1.43 | 0.13 | 1.23 | 482 |
| | Arendsee | -1.06 | 1.46 | 0.11 | 1.13 | 445 |
| $T_b$ | Müggelsee | -1.72 | 1.94 | 0.15 | 1.13 | 1303 |
| | Heiligensee | -1.22 | 1.46 | 0.17 | 0.95 | 383 |
| | Stechlinsee | -0.29 | 0.62 | 0.13 | 1.93 | 482 |
| | Arendsee | 0.22 | 0.95 | 0.19 | 2.45 | 444 |
| $h_{mix}$ | Heiligensee | -1.56 | 0.95 | 0.43 | 0.85 | 113 |
| | Stechlinsee | 1.20 | 1.55 | 0.28 | 0.66 | 191 |
| | Arendsee | -2.1 | 2.0 | 0.42 | 0.36 | 134 |
| Stratification duration | Heiligensee | -35.3 | 17.2 | 0.16 | 0.60 | 21 |
| | Stechlinsee | -39.4 | 15.5 | 0.17 | 0.73 | 33 |
| | Arendsee | -21.2 | 12.0 | 0.11 | 0.74 | 19 |
| Stratification begin | Heiligensee | 8.6 | 10.8 | 0.14 | 0.52 | 21 |
| | Stechlinsee | 16.9 | 10.2 | 0.20 | 0.63 | 22 |
| | Arendsee | 5.8 | 8.7 | 0.10 | 0.93 | 24 |
| Stratification end | Heiligensee | -1.9 | 16.3 | 0.06 | 0.69 | 20 |
| | Stechlinsee | -18.5 | 9.6 | 0.06 | 0.58 | 21 |
| | Arendsee | -15.3 | 11.5 | 0.06 | 0.30 | 21 |



**Table 3: Projected trends (2006 – 2100) in lake temperature and stratification characteristics, with 95% confidence intervals in parentheses. doy = day of the year; ns = not significant; italics denote trends with 0.01 < p < 0.05; all other trends are significant at p < 0.001.**

| | Unit decade$^{-1}$ | Müggelsee | Heiligensee | Stechlinsee | Arendsee |
|---|---|---|---|---|---|
| **Surface temperature** | | | | | |
| Annual | °C | 0.12 (0.11 – 0.14) | 0.13 (0.11 – 0.14) | 0.14 (0.12 – 0.15) | 0.14 (0.13 – 0.16) |
| Winter | °C | 0.25 (0.23 – 0.27) | 0.26 (0.24 – 0.28) | 0.21 (0.19 – 0.23) | 0.20 (0.18 – 0.23) |
| Spring | °C | 0.22 (0.19 – 0.24) | 0.23 (0.20 – 0.25) | 0.30 (0.28 – 0.33) | 0.31 (0.29 – 0.34) |
| Summer | °C | ns | ns | *0.02* (0.00 – 0.05) | ns |
| Autumn | °C | *0.02* (0.00 – 0.05) | *0.02* (0.00 – 0.04) | ns | 0.03 (0.01 – 0.04) |
| Maximum | °C | 0.13 (0.09 – 0.17) | 0.12 (0.08 – 0.16) | 0.14 (0.11 – 0.17) | 0.13 (0.10 – 0.16) |
| Timing of maximum | doy | -1.0 (-1.4 – -0.6) | -0.8 (-1.2 – -0.4) | -1.1 (-1.5 – -0.7) | -0.9 (-1.3 – -0.6) |
| **Bottom temperature** | | | | | |
| Annual | °C | 0.10 (0.09 – 0.12) | 0.10 (0.08 – 0.11) | 0.07 (0.06 – 0.08) | 0.08 (0.07 – 0.10) |
| Winter | °C | 0.20 (0.18 – 0.22) | 0.18 (0.16 – 0.20) | 0.12 (0.10 – 0.13) | 0.11 (0.10 – 0.13) |
| Spring | °C | 0.17 (0.15 – 0.20) | 0.12 (0.09 – 0.15) | 0.06 (0.05 – 0.08) | 0.08 (0.07 – 0.10) |
| Summer | °C | ns | 0.06 (0.04 – 0.09) | 0.05 (0.04 – 0.07) | 0.09 (0.08 – 0.10) |
| Autumn | °C | *0.02* (0.002 – 0.04) | *0.02* (0.00 – 0.04) | 0.04 (0.03 – 0.06) | 0.06 (0.04 – 0.07) |
| **Summer stratification** | | | | | |
| Duration | d | 0.8 (0.4 – 1.3) | 0.9 (0.5 – 1.2) | 1.8 (1.4 – 2.1) | 1.6 (1.3 – 1.9) |
| Start timing | doy | -1.5 (-1.9 – -1.0) | -1.5 (-1.8 – -1.2) | -2.1 (-2.4 – -1.9) | -2.2 (-2.5 – -2.0) |
| End timing | doy | ns | -0.7 (-0.9 – -0.4) | -0.4 (-0.6 – -0.2) | -0.7 (-0.9 – -0.5) |
| Summer mixed layer depth | m | 0.02 (0.01 – 0.02) | 0.03 (0.02 – 0.04) | ns | 0.04 (0.03 – 0.04) |
| **Winter stratification** | | | | | |
| Duration | d | -0.9 (-1.2 – -0.6) | -1.1 (-1.4 – -0.8) | -1.7 (-2.2 – -1.3) | -2.0 (-2.5 – -1.6) |
| Start timing | doy | -0.8 (-1.4 – -0.3) | -1.0 (-1.5 – -0.5) | ns | *-0.7* (-1.2 – -0.1) |
| End timing | doy | -1.2 (-1.8 – -0.6) | -1.4 (-1.9 – -0.8) | -1.6 (-2.2 – -1.1) | -2.3 (-3.0 – -1.6) |
| **Ice cover** | | | | | |
| Duration | d | -0.9 (-1.1 – -0.6) | -1.0 (-1.3 – -0.7) | -0.4 (-0.5 – -0.3) | -0.2 (-0.3 – -0.1) |
| First freeze | doy | ns | ns | ns | ns |
| Last thaw | doy | -1.4 (-2.1 – -0.6) | -1.4 (-2.1 – -0.7) | -1.8 (-2.8 – -0.9) | ns |

