# Peer review of "Future projections of temperature and mixing regime of European temperate lakes"

_Hydrology and Earth System Sciences, 2018_

## Referee Comment (RC1) · Anonymous Referee #1 · 28 Dec 2018

Shatwell et al., "Future projections of temperature and mixing regime of European temperate lakes."

General comments

In the present paper the authors examine the response of 4 European lakes to global warming. The study is based on numerical simulations where RCP45 climate scenarios of several global models are used as lateral boundary conditions to the RCA4 regional climate model. This ensemble is then used to drive the one-dimensional lake model FLake simulations and study the evolution of surface temperature, ice cover and wind-mixing regimes. The authors demonstrate a warming tendency of the selected lakes and a change in mixing regimes. They also highlight the key role of transparency and the seasonal link with stratification and mixing.

[Figure]

The manuscript is well structured and written, the demonstrations are satisfactorily conducted and the discussions of interest. However, there are some specific questions I would like to be answered before accepting the manuscript for publication.

Specific comments

The title does not completely reflect the content of the manuscript. As stated by the authors it turns out to be very difficult to generalize their conclusions due to the wide variability of lakes and behaviors. European temperate zones extend from the Atlantic Ocean to the Black Sea and are bordered to the north by the boreal zone and to the south by the Mediterranean. The impact of climate change may vary substantially in such a wide region. It would be more appropriate to limit the scope of their conclusions to the 4 studied lakes. I suggest to modify the title accordingly.

Global climate scenarios are used as is without considering bias correction of any kind. Even if this is probably true that the way how to correct climate simulations is debatable, the climate response of individual lakes is mainly driven by air temperature. RCA4, like the other regional climate models is biased in temperature (Strandberg et al. (2014): CORDEX scenarios for Europe from the Rossby Centre regional climate model RCA4). How does RCA4 temperature bias affect the conclusion of the current study?

Precipitation is not mentioned as input to FLake model. However, in winter snow can reinforce substantially lake insulation in presence of ice. In RCA4 precipitation is also biased. Is the snow module activated in FLake? Please add a comment on that particular point and discuss how snow could modulate the conclusions of the paper, at least in the close future before the warming prevents ice formation.

Flake is calibrated with ERA-Interim data, it is not clear however which calibration period was used: figure 3 indicates 1996-2002 but this is not explicitly stated in the paper.

Calibration of FLake parameters allow correcting biases in ERA-Interim forcing (Biases due to daily variables, to sub-daily interpolation, etc.). For the future period 2020-2100,

it is not proven that these calibrations are best when forcing is made with RCA4. I encourage authors to compare RCA4 model runs for present period with and without calibration, using ERA-Interim as lateral boundary conditions, and discuss the impact in terms of surface temperature, icing, wind-mixing regimes, etc.

In Müggelsee a specific calibration is performed to account for a water supply from a connected river. Is there any signal in climate simulations that confirms this river discharge will be as important as in the present climate? A smaller discharge in the future (due to less precipitation, more evaporation, etc.) would for instance impact transparency and change the calibration results.

In figure 9 only the 2050-2059 period is considered? Are the results also valid for the other time periods? Please add a comment on that point page 10.

The ensemble of 12 members is not discussed in terms of dispersion: a rank diagram of air temperature is probably very important to discuss the ensemble model dispersion and demonstrate this ensemble is enough-dispersive to represent the climate variability.

Sine data are used to reconstruct wind which is a key variable for the mixing-regime of lakes. It is not clearly proven how accurate wind reconstruction is and how it compares to Potsdam dataset. Please indicate Potsdam location in Figure 1.

Temporal downscaling of humidity is performed linearly. Is it relative humidity that is considered as input to FLake? Usually specific humidity is used. Please clarify.

In the presentation of FLake model runs, it would have been helpful to clearly explain which time step was used: sub-daily variables are constructed but then is the atmospheric forcing 6-hourly, 3-hourly, etc.?

Technical comments

Page 9 line 9: ensemble

Page 12 line 25: transparency

Page 12 line 28: Heiligensee

---

## Referee Comment (RC2) · Anonymous Referee #2 · 22 Jan 2019

Future projections of temperature and mixing regime of European temperate lakes by Tom Shatwell, Wim Thiery and Georgiy Kirillin is dedicated to qualify the role of individual lake characteristics, like water temperature, vertical mixing, ice formation, and water transparency, in their response to regionally homogeneous 2 meter air temperature warming. This study is designed to analyse the effects of seasonality on the response of northern temperate lakes to projected future warming. Changes in lake mixing regime reflect climate change impact on lakes, understanding how the seasonality of warming interacts with lake stratification and ice cover is vital to interpret differences in future lake warming trends. Four intensively studied German lakes of varying morphology and mixing regime (two shallow and two deep lakes, where each pair is similar in terms of morphology but different in terms of water transparency and/or

mixing regime) situated within a distance of $\leq$ 150 km from each other were modelled with a one-dimensional lake model FLake, which is based on a two-layer parametric representation of the vertical temperature structure. The upper layer is treated as well-mixed and vertically homogeneous. The structure of the lower stably-stratified layer (lake thermocline), upper layer of the bottom sediments and the ice cover are parameterized using a self-similar representation of the temperature profile. For each lake FLake model was forced with an ensemble of 12 climate projections (RCP4.5) from 2006 up to 2100. The ensemble was assembled from different downscaled global climate models (MPI-ESM-LR, EC-EARTH, CanESM2, CNRM-CM5, CSIRO-Mk3-6-0, GFDL-ESM2M, MIROC5, NorESM1-M) each providing lateral boundary conditions to the RCA4 regional climate model. All simulations were performed at a horizontal resolution of 0.44°. Main results are: (i) lakes warming at $0.10 - 0.11$ °C decade-1, which is $75 - 90\%$ of the projected air temperature trend; (ii) advanced ice thaw and summer stratification by 1.5–2.2 and 1.4–1.8 d decade-1 respectively, less sensitivity of autumn turnover and winter freeze timing; (iii) summer mixed layer depth not affected by warming but sensitive to changes in water transparency; (iv) transparency decrease dampens the effect of warming on mean temperature but amplifies its effect on stratification; (v) heat store and lake respond to climate warming is determined not only by lake morphology, but also by mixing regime. Altogether this study suggests that warming over the next century will gradually shift many temperate dimictic lakes towards a predominantly monomictic regime, particularly since projected warming rates are highest in winter and spring, with deeper lakes shifting before shallower ones. On the other hand, shifts from polymixis to dimixis are more likely to occur due to a change in transparency or depth than due to climate warming alone.

General comments Paper addresses relevant scientific questions within the scope of HESS, namely internal physical mechanisms determining the response of lakes to a future warmer climate. It presents new analysis of northern temperate lakes variables in a projected moderate climate warming scenario (Radiative Concentration Pathway 4.5, RCP4.5). In addition, all lakes used in this study have different combination of

morphology and mixing regime, yet they all are situated rather close to each other, what makes analysis even more interesting and relevant. Paper reaches substantial conclusions on lake vertical mixing, ice formation dates and water transparency behaviour according to the projected climate change. Methods and assumptions are valid and rather clearly outlined, the only clarification is needed for light extinction constant for Arendsee. Paper results are sufficient to support the interpretations and conclusions presented. Model experiment description and explanations of result calculation methodology are sufficiently complete and precise to allow their reproduction by fellow scientists (good traceability of results). Also, authors give possibility to download initial data or model output. They give proper credit to related work and clearly indicate their own new/original contribution to the analysis of lake main variables in future warming climate and indicate each authors input. Paper title clearly reflects the contents of the manuscript, abstract provides a concise and complete summary of the research done. Overall presentation of the paper is well structured and clear, language fluent and precise, all mathematical formulae, symbols, abbreviations, and units are correctly defined and used, number and quality of references are appropriate.

Specific comments Although paper gives an impression of a proper well-written, well-referenced and well-structured manuscript, I think that following clarifications/additions should be done prior publication: 1. p4, l13 - could you add some explanation how constant 2.17 was derived? 2. p6, l5 - could you specify on the technique used to detect lake variable changes for rather small lakes (lake water surface area vary between 0.3-7.3 km2) if simulations were performed at a horizontal resolution 0.44°? Or this is only atmospheric forcing resolution? 3. p6, l17 - what period of data was used to empirically determine the value? 4. p27, Fig.5 - Stechlinsee and Arendsee patterns look very similar, horizontal and vertical grids would help to better see if any difference is present. 5. p28, Fig.6 - mixed layer depth values especially for Stechlinsee and Arendsee are not visible (as well as winter and autumn periods for all 4 lakes), as it was mentioned that there are 58% and 75% respectively that these lakes are ice-free is it possible to show values on the graph (extending y and x axis)? Or an explanation

why it shouldn't be done? 6. p30, Fig.8 - could you explain an interesting behaviour pattern of Heiligensee in annual mean temperature graph? 7. p32, Table1 - could extra line with ice duration in days be added? 8. p33, Table2 - some correction with table rows is needed.

---

## Author Comment (AC1) · 11 Feb 2019

We thank referee #1 for her/his time and effort in formulating helpful comments.

Comment: The title does not completely reflect the content of the manuscript.

Answer: We respectfully disagree. The title reflects exactly what the reader will find in the paper. All the lakes considered in the study are European temperate lakes, located in the area not affected by marginal influence of neighbouring climate zones, whether Mediterranean, alpine, or boreal. Moreover, the four lakes are all representatives of the major lake seasonal mixing types found in temperate climates providing the modeling results with generality. Citing Reviewer2, who explicitly underlined the background idea of our study setup, "...all lakes used in this study have different combination of morphol-

ogy and mixing regime, yet they are situated rather close to each other, what makes analysis even more interesting and relevant...Paper title clearly reflects the contents of the manuscript...".

Comment: Global climate scenarios are used as is without considering bias correction of any kind. How does RCA4 temperature bias affect the conclusion of the current study?

Answer: See our reply to the specific comment on bias below.

Comment: Precipitation is not mentioned as input to FLake model. However, in winter snow can reinforce substantially lake insulation in presence of ice. In RCA4 precipitation is also biased. Is the snow module activated in FLake? Please add a comment on that particular point and discuss how snow could modulate the conclusions of the paper, at least in the close future before the warming prevents ice formation

Answer: Precipitation did not enter the model directly, neither was the snow included into the ice modeling. There are several indications suggesting that including the snow as model input would not improve the predictive value of the model outcomes: (i) The current version of FLake treats the thermal regime under ice in a simplified way, without taking into account the short-wave radiation penetrating into the water column under the ice cover. In that sense, adding the snow as an insulation for the radiation flux does not change the mixing physics of the ice-covered period much. (ii) In temperate regions, the relatively short ice-covered periods on lakes are weakly affected by the snow cover compared to e.g. boreal and arctic lakes. This fact was also supported by the study of the FLake performance for the ice modeling on Lake Müggelsee (Bernhardt et a. 2012). Taking also into account the additional uncertainty related to the absense of exact information on snow proportion, the complex snow and ice rheology at temperatures close to the melting point, and the general shortening of the ice cover period in the future scenarios, an inclusion of precipitation into scenarios did not seem reasonable. /a version of this passage can be included into the paper for clarity/

Comment: Flake is calibrated with ERA-Interim data, it is not clear however which calibration period was used: figure 3 indicates 1996-2002 but this is not explicitly stated in the paper.

Answer: We assessed model performance using temperature profiles in the period from 1979 to 2014 in Müggelsee (measured weekly, and from 2004 to 2014 hourly to assess short-term stratification), from 1979 – 2001 in Heiligensee (monthly), from 1991 to 2012 in Stechlinsee (weekly to monthly), and from 1979 to 2010 in Arendsee (weekly to monthly). This sentence can be added to the revised manuscript.

Comment: Calibration of FLake parameters allow correcting biases in ERA-Interim forcing (Biases due to daily variables, to sub-daily interpolation, etc.). For the future period 2020-2100, it is not proven that these calibrations are best when forcing is made with RCA4. I encourage authors to compare RCA4 model runs for present period with and without calibration, using ERA-Interim as lateral boundary conditions, and discuss the impact in terms of surface temperature, icing, wind-mixing regimes, etc.

Answer: This is of course true, however, we not only used the RCA4 model in the ensemble, but a total of different 5 regional climate models (CCLM4-8-17, REMO2009, HIRHAM5, RACMO22E, RCA4), each with their own bias. In fact, each of the 12 GCM-RCM combinations has its own bias, which would imply 12 different calibrations to account for both RCM and GCM bias. We chose to use ERA-Interim here so that we could work with only one parameter set. Nevertheless, the potential bias is a good point, so we reran the model (with the parameterization used for the manuscript) with the historical hindcast of each of the 12 GCM-RCM combinations in the ensemble, and calculated the bias of the key variables (temperature, mixing etc - see below) we analyzed, as the reviewer suggested. The mean bias of each variable forced by the ensemble hindcast barely differs from the bias obtained using ERA-Interim. With the parameterization we used, the ensemble showed on average a smaller absolute bias in bottom temperature, mixed layer depth, stratification duration and stratification onset timing and a slightly greater bias only in surface temperature and stratification

breakdown timing. This provides high confidence that our results and conclusions are relatively unaffected by bias. We will include this information and discussion of the consequences in a revised manuscript.

Bias for key lake variables with the model (forced by ERA-Interim / the ensemble, same model parameters).

$T_s$ (degrees C): Müggelsee: -0.67 / -0.83, Heiligensee: -1.07 / -0.82, Stechlinsee: -1.83 / -2.21, Arendsee: -1.06 / -1.39

$T_b$ (degrees C): Müggelsee: -1.72 / -2.14, Heiligensee: -1.22 / 0.08, Stechlinsee: -0.29 / 0.06, Arendsee: 0.22 / 0.31

$h_{mix}$ (m): Heiligensee: -1.56 / -1.12, Stechlinsee: 1.20 / -0.13, Arendsee: -2.1 / -0.85

Stratification duration (d): Heiligensee: -35.3 / -20.9, Stechlinsee: -39.4 / -37.1, Arendsee: -21.2 / -18.1

Stratification begin (day of year): Heiligensee: 8.6 / 13.5, Stechlinsee: 16.9 / 9.3, Arendsee: 5.8 / -0.1

Stratification end (day of year): Heiligensee: -1.9 / -20.6, Stechlinsee: -18.5 / -28.3, Arendsee: -15.3 / -18.4

Comment: In Müggelsee a specific calibration is performed to account for a water supply from a connected river. Is there any signal in climate simulations that confirms this river discharge will be as important as in the present climate? A smaller discharge in the future (due to less precipitation, more evaporation, etc.) would for instance impact transparency and change the calibration results.

Answer: Our scenarios assume a conservative behavior with respect to the river discharge. While effects of the changed river flow on the temperature and mixing regime can indeed be manifold, their assessment requires more detailed information than a simple scenario for an average discharge. The atmospheric influence remains however a primary factor for the physical processes in lakes, hence the focus of model scenarios on the lake-atmosphere interaction. Moreover, as the reviewer suggested, another important factor is transparency. However, it is unclear how this will change, with reports in the literature that it is likely to both increase and decrease. It is linked to runoff, carbon export and nutrient loading. For this reason we performed the sensitivity analysis with transparency, and found that changes in transparency near the current values are unlikely to substantially alter the thermal response of the lakes to warming.

Comment: In figure 9 only the 2050-2059 period is considered? Are the results also valid for the other time periods? Please add a comment on that point page 10.

Answer: Yes the results are valid for those periods too in the sense that the pattern and relationship to extinction is the same (compare Fig 1 and Fig 2). The absolute values of stratification duration of course change somewhat with warming. We added a comment in this regard to the revised manuscript.

Comment: The ensemble of 12 members is not discussed in terms of dispersion: a rank diagram of air temperature is probably very important to discuss the ensemble model dispersion and demonstrate this ensemble is enough-dispersive to represent the climate variability.

Answer: We created rank (cumulative distribution) diagrams of the monthly and annual mean temperatures of the historical scenarios of the ensemble and compared these with the corresponding temperatures of the ERA-Interim reanalysis and the actual measured air temperatures at the Menz weather station for the period 1991 to 2010 (Fig. 3). The figure shows that the dispersion of the ensemble is quite comparable to that of both the ERA-Interim and the observed data. The ensemble tended to slightly underestimate the frequency of extremely warm months in the upper 10th percentile (a) but not the frequency of warm years (b). This may affect our estimates of peak summer temperatures in all lakes and the frequency of extended stratification events during heatwaves in polymictic Müggelsee. However, it should not influence our overall

conclusions. This will be included in a revision.

Comment: Sine data are used to reconstruct wind which is a key variable for the mixing-regime of lakes. It is not clearly proven how accurate wind reconstruction is and how it compares to Potsdam dataset. Please indicate Potsdam location in Figure 1.

Answer: The algorithm for generating the subdaily data accurately reproduced the complexity of the observed windspeed dynamics, as shown in Fig. 4. It produced realistic behavior of day-to-day windspeed (Fig 4a), as well as the hourly variation of mean windspeed and associated variability (Fig 4b, c), and also the seasonal change of this hourly variation (Fig 4b, c), while still preserving the given daily mean windspeeds (Fig 4d). We mistakenly stated that the algorithm was based on Potsdam weather data. In fact it was based on the weather data from the Menz station, located 5 km from the shore of Stechlinsee, about 100 km north of Potsdam. The map in Manuscript-Fig 1 is unable to resolve this small distance, so we added it in the text.

Comment: Temporal downscaling of humidity is performed linearly. Is it relative humidity that is considered as input to FLake? Usually specific humidity is used. Please clarify.

Answer: Yes Flake uses specific humidity as input. We tested this interpolation approach against the observed weather data at the Menz station. Assuming linear day-to-day variation of specific humidity, we calculated the diurnal variation of relative humidity based on the diurnal variation of air temperature used in our approach. This disaggregated relative humidity matched the diurnal variation of observed relative humidity very closely.

Comment: In the presentation of FLake model runs, it would have been helpful to clearly explain which time step was used: sub-daily variables are constructed but then is the atmospheric forcing 6-hourly, 3-hourly, etc.?

Answer: True, we used 6-hourly atmospheric forcing, constructed from the daily values.

[Figure]

The sentence now reads: "These variables were available at daily resolution in the climate projections, and were downscaled for model simulations to 6-hourly resolution with the same daily mean to account for diurnal forcing."

Comment: Page 9 line 9: ensemble; Page 12 line 25: transparency; Page 12 line 28: Heiligensee

Answer: Ensemble is corrected, thanks, but transparency and Heiligensee seem okay?

Reference: Bernhardt, J., et al. (2012). "Lake ice phenology in Berlin-Brandenburg from 1947-2007: observations and model hindcasts." Climatic Change 112(3-4): 791-817.
* * *
[Figure]

**Fig. 1.** Effect of averaging period on Manuscript-Fig 7: the relation for the averaging period 2010-2019.

[Figure]

**Fig. 2.** As for Fig 1 but calculated with averaging period 2090-2099.

[Figure]

**Fig. 3.** Cumulative distribution function of centred monthly (a) and annual mean air temperature (b), from 1991 - 2010 data. Grey lines: ensemble, green lines: ERA-Interim, blue lines: Menz station

[Figure]

**Fig. 4.** Validation of algorithm for disaggregating daily windspeeds (black) against Menz observed data (blue). a) data sample, b,c) averaged hourly mean and sd in Feb and July, d) mean (re-aggregated) vs obs

---

## Author Comment (AC2) · 11 Feb 2019

We thank referee #2 for her/his time and effort and for providing constructive comments.

General comments Comment: Paper addresses relevant scientific questions within the scope of HESS, namely internal physical mechanisms determining the response of lakes to a future warmer climate. It presents new analysis of northern temperate lakes variables in a projected moderate climate warming scenario (Radiative Concentration Pathway 4.5, RCP4.5). In addition, all lakes used in this study have different combination of morphology and mixing regime, yet they all are situated rather close to each other, what makes analysis even more interesting and relevant. Paper reaches substantial conclusions on lake vertical mixing, ice formation dates and water transparency behaviour according to the projected climate change. Methods and assumptions are valid and rather clearly outlined, the only clarification is needed for light extinction constant for Arendsee. Paper results are sufficient to support the interpretations and conclusions presented. Model experiment description and explanations of result calculation methodology are sufficiently complete and precise to allow their reproduction by fellow scientists (good traceability of results). Also, authors give possibility to download initial data or model output. They give proper credit to related work and clearly indicate their own new/original contribution to the analysis of lake main variables in future warming climate and indicate each authors input. Paper title clearly reflects the contents of the manuscript, abstract provides a concise and complete summary of the research done. Overall presentation of the paper is well structured and clear, language fluent and precise, all mathematical formulae, symbols, abbreviations, and units are correctly defined and used, number and quality of references are appropriate.

Answer: Thank you for the overview and highlighting the strengths of the manuscript. Our response to the extinction coefficient is given below.

Specific comments Comment: 1. p4, l13 - could you add some explanation how constant 2.17 was derived?

Answer: Light extinction ($\gamma$) in Müggelsee, Heiligensee, and Stechlinsee was calculated using the Lambert-Beer law from simultaneous light measurements at different depths (generally 0.5 m apart) recorded with spherical sensors (Licor, Nebraska). Using regression, we related light extinction to parallel measurements of Secchi depth using the relationship $\gamma$ = c / hsecchi (Poole and Atkins, 1929). We determined the constant c to be 2.05±0.04 (mean ± s.e, n=300) for Müggelsee, 2.13±0.10 (n=52) for Heiligensee (Shatwell et al., 2016), and 2.33±0.08 (n=57) for Stechlinsee. In the absence of direct measurements in Arendsee, light extinction was estimated from Secchi depth measurements as $\gamma$ = 2.17 / hsecchi, where the constant c = 2.17 was simply the mean of the estimates from the other three lakes.

Comment: 2. p6, l5 - could you specify on the technique used to detect lake variable changes for rather small lakes (lake water surface area vary between 0.3- 7.3 km2) if simulations were performed at a horizontal resolution 0.44_? Or this is only atmospheric forcing resolution?

Answer: This was the resolution of the atmospheric forcing. We will clarify this point in the revision.

Comment: 3. p6, l17 - what period of data was used to empirically determine the value?

Answer: We used the period 1.11.1991 to 24.8.2004. However we mistakenly stated the name of the weather station, which was Menz, not Potsdam.

Comment: 4. p27, Fig.5 - Stechlinsee and Arendsee patterns look very similar, horizontal and vertical grids would help to better see if any difference is present.

Answer: we added gridlines to the plots and slightly expanded the vertical scale (see Fig. 1).

Comment: 5. p28, Fig.6 - mixed layer depth values especially for Stechlinsee and Arendsee are not visible (as well as winter and autumn periods for all 4 lakes), as it was mentioned that there are 58% and 75% respectively that these lakes are ice-free is it possible to show values on the graph (extending y and x axis)? Or an explanation why it shouldn't be done?

Answer: We intentionally chose the scales to focus on the stable summer stratification period. Since weak stratification forms and breaks down regularly in spring, and the mixed layer depth jumps from shallow to deep before the more stable seasonal stratification begins, showing the ensemble mean mixed depth is not meaningful in early spring. Important information here was presented in the form of stratification start and end dates and trends in Fig 7 and Table 3. Thus we decided in this case not to alter the figure as suggested but retain the focus on the stable summer period.

Comment: 6. p30, Fig.8 - could you explain an interesting behaviour pattern of Heiligensee in annual mean temperature graph?

Answer: Our hypothesis for this behavior is: When extinction is very low, not all of the incident radiation is absorbed in the water column in Müggelsee because it is relatively shallow. Here an increase in extinction causes the mean temperature to increase because more radiation is absorbed in the water column. Stechlinsee and Arendsee on the other hand are deep enough that all radiation is absorbed, even at extremely low extinction. Here an increase in extinction causes the mean temperature to decrease because less radiation penetrates to deeper waters. Heiligensee is at the transition, also shifting from polymictic to dimictic with increasing extinction. This is apparently the cause of the interesting behavior, possibly with an interaction with warmer air temperature projected at the end of the century. This explanation will be added to the revised manuscript.

Comment: 7. p32, Table1 - could extra line with ice duration in days be added?

Answer: We added ice duration statistics for the two lakes for which we had ice data (Müggelsee and Stechlinsee - see table in the attached supplement).

Comment: 8. p33, Table2 - some correction with table rows is needed.

Answer: Thanks for the info, this seems to be a formatting issue. The section titles are too long to fit on one line in the narrow column, so the first row of each subsection has 2 lines. If the manuscript is accepted, this problem should disappear during typesetting.

Please also note the supplement to this comment:
https://www.hydrol-earth-syst-sci-discuss.net/hess-2018-588/hess-2018-588-AC2-supplement.pdf
* * *
Müggelsee

Heiligensee

Stechlinsee

Arendsee

Temperature (°C)

20
10
0

20

10

0

20

10

0

20

10

0

1996    1997    1998    1999    2000    2001    2002

**Fig. 1.** Revised version of manuscript Fig 5 - validation of the model against measured data in the 4 lakes.

**Supplement:**

**Goodness of fit statistics of the four study lakes. Stratification statistics not shown for polymictic Müggelsee.**

| | Lake | Bias | $RMSE_c$ | $RMSE_n$ | $\sigma_n$ | $n$ |
|---|---|---|---|---|---|---|
| $T_s$ | Müggelsee | -0.67 | 1.02 | 0.07 | 1.13 | 1335 |
| | Heiligensee | -1.07 | 1.04 | 0.09 | 1.18 | 383 |
| | Stechlinsee | -1.83 | 1.43 | 0.13 | 1.23 | 482 |
| | Arendsee | -1.06 | 1.46 | 0.11 | 1.13 | 445 |
| $T_b$ | Müggelsee | -1.72 | 1.94 | 0.15 | 1.13 | 1303 |
| | Heiligensee | -1.22 | 1.46 | 0.17 | 0.95 | 383 |
| | Stechlinsee | -0.29 | 0.62 | 0.13 | 1.93 | 482 |
| | Arendsee | 0.22 | 0.95 | 0.19 | 2.45 | 444 |
| $h_{mix}$ | Heiligensee | -1.56 | 0.95 | 0.43 | 0.85 | 113 |
| | Stechlinsee | 1.20 | 1.55 | 0.28 | 0.66 | 191 |
| | Arendsee | -2.1 | 2.0 | 0.42 | 0.36 | 134 |
| Stratification duration | Heiligensee | -35.3 | 17.2 | 0.16 | 0.60 | 21 |
| | Stechlinsee | -39.4 | 15.5 | 0.17 | 0.73 | 33 |
| | Arendsee | -21.2 | 12.0 | 0.11 | 0.74 | 19 |
| Stratification begin | Heiligensee | 8.6 | 10.8 | 0.14 | 0.52 | 21 |
| | Stechlinsee | 16.9 | 10.2 | 0.20 | 0.63 | 22 |
| | Arendsee | 5.8 | 8.7 | 0.10 | 0.93 | 24 |
| Stratification end | Heiligensee | -1.9 | 16.3 | 0.06 | 0.69 | 20 |
| | Stechlinsee | -18.5 | 9.6 | 0.06 | 0.58 | 21 |
| | Arendsee | -15.3 | 11.5 | 0.06 | 0.30 | 21 |
| Ice duration | Müggelsee | -29.9 | 22.8 | 0.64 | 0.14 | 23 |
| | Stechlinsee | -21.1 | 24.7 | 0.81 | 0.07 | 13 |

---

## Author Response (AR1)

**Revisions in response to comments of Referee #1**

We thank referee #1 for the helpful comments.

**Comment:** The title does not completely reflect the content of the manuscript.

**Answer**: As explained in our response to the reviews, we felt that the title is appropriate and retained the original title.

**Comment:** Global climate scenarios are used as is without considering bias correction of any kind. How does RCA4 temperature bias affect the conclusion of the current study?

**Answer**: See our reply to the specific comment on bias below.

10   **Comment:** Precipitation is not mentioned as input to FLake model. However, in winter snow can reinforce substantially lake insulation in presence of ice. In RCA4 precipitation is also biased. Is the snow module activated in FLake? Please add a comment on that particular point and discuss how snow could modulate the conclusions of the paper, at least in the close future before the warming prevents ice formation

**Answer**: We added the following text to section 2.7 in the methods:

15   "Precipitation, including snow cover on ice, were not included in modelling. There are several indications suggesting that including the snow as model input would not improve the predictive value of the model outcomes. Firstly, the current version of FLake treats the thermal regime under ice in a simplified way, without taking into account the short-wave radiation penetrating into the water column under the ice cover. Therefore including the insulating effect of snow on the radiation flux does not substantially change the mixing physics of the ice-covered period. Secondly, in temperate regions, the
20   relatively short ice-covered periods on lakes are weakly affected by the snow cover compared to e.g. boreal and arctic lakes. This fact was also supported by the study of the FLake performance for the ice modeling on Lake Müggelsee (Bernhardt et al., 2012)."

**Comment:** Flake is calibrated with ERA-Interim data, it is not clear however which calibration period was used: figure 3
25   indicates 1996-2002 but this is not explicitly stated in the paper.

**Answer**: We added the following sentence to section 2.7 in the methods:

"Here we assessed model performance using temperature profiles in the period from 1979 to 2014 in Müggelsee (measured weekly, and from 2004 to 2014 hourly to assess short-term stratification), from 1979 – 2001 in Heiligensee (monthly), from 1991 to 2012 in Stechlinsee (weekly to monthly), and from 1979 to 2010 in Arendsee (weekly to monthly)."

**Comment:** Calibration of FLake parameters allow correcting biases in ERA-Interim forcing (Biases due to daily variables, to sub-daily interpolation, etc.). For the future period 2020-2100, it is not proven that these calibrations are best when forcing is made with RCA4. I encourage authors to compare RCA4 model runs for present period with and without calibration, using

ERA-Interim as lateral boundary conditions, and discuss the impact in terms of surface temperature, icing, wind-mixing regimes, etc.

**Answer:** We followed the reviewer's suggestion and reran the model. We added the following text to section 2.7 of the methods:

5 "Using a model calibrated on a different dataset (ERA-Interim) than the climate projections can introduce bias into the results. Thus we reran the model calibrated on ERA-Interim data with forcing from the historical hindcasts of each of the 12 GCM-RCM combinations in the ensemble. The mean bias of each variable forced by the ensemble hindcast barely differed from the bias obtained using ERA-Interim forcing. In fact, the ensemble showed on average a smaller absolute bias in bottom temperature, mixed layer depth, stratification duration and stratification onset timing and a slightly greater bias only

10 in surface temperature and stratification breakdown timing. Therefore, our simulations are largely unaffected by bias."

We also added the following text to section 4.6 in the discussion:

"Additional error can arise through bias in the forcing meteorological data but we found that the mean bias in key variables obtained from the ensemble forcing did not differ from the bias of the model calibrated with ERA-Interim reanalysis data."

15 **Comment:** In Müggelsee a specific calibration is performed to account for a water supply from a connected river. Is there any signal in climate simulations that confirms this river discharge will be as important as in the present climate? A smaller discharge in the future (due to less precipitation, more evaporation, etc.) would for instance impact transparency and change the calibration results.

**Answer:** We added the following text to section 2.7 in the methods:

20 "Moreover, we did not account for changes in inflow because our focus was on lake-atmosphere interactions. Instead we assessed the sensitivity of the lakes to a change in transparency, which can be expected if warming alters runoff as well as nutrient and carbon export."

**Comment:** In figure 9 only the 2050-2059 period is considered? Are the results also valid for the other time periods? Please

25 add a comment on that point page 10.

**Answer:** We added the following text to section 3.4 in the results:

"Fig. 11 was derived for the period 2090-2099. Although stratification duration increased over time as indicated in Table 3, the relative dependence on extinction was similar during the whole period covered in the ensemble."

Note that Figure 9 became Figure 11 in the revision. Also, in the revision, we decided is was more relevant to show the

30 period at the end of the century 2090-2099 rather than 2050-2059.

**Comment:** The ensemble of 12 members is not discussed in terms of dispersion: a rank diagram of air temperature is probably very important to discuss the ensemble model dispersion and demonstrate this ensemble is enough-dispersive to represent the climate variability.

**Answer**: We added a new figure (Fig. 2) with rank (cumulative distribution) diagrams of the monthly and annual mean temperatures of the historical scenarios of the ensemble compared with the corresponding temperatures of the ERA-Interim reanalysis and the actual measured air temperatures at the Menz weather station for the period 1991 to 2010.

We also added the following text to section 2.4 in the methods:

5  "The dispersion of mean monthly and annual air temperature in the ensemble is quite comparable to that of both the ERA-Interim reanalysis dataset and the observed data from a nearby weather station (Fig. 2). The ensemble tended to slightly underestimate the frequency of extremely warm months in the upper 10th percentile (Fig. 2a) but not the frequency of warm years (Fig. 2b)."

We also added the following texts to section 4.2 and 4.6 in the discussion:

10  "The model may have slightly underestimated the increase in summer peak temperatures and occurrence of extended stratification during heatwaves in polymictic Müggelsee because the ensemble tended to slightly underestimate the frequency of extremely warm months."

"Another source of error can be incorrect representation of dispersion in the forcing ensemble. However, here we found that the dispersion of air temperature was quite similar to the dispersion in measured meteorological data and the ERA-Interim

15  reanalysis. The slight underestimation of extremely warm periods in the ensemble may also cause a slight underestimation in our results of peak summer temperatures in all lakes and of the frequency of extended stratification events during heatwaves in polymictic Müggelsee. This should not influence our overall conclusions."

**Comment:** Sine data are used to reconstruct wind which is a key variable for the mixing-regime of lakes. It is not clearly

20  proven how accurate wind reconstruction is and how it compares to Potsdam dataset. Please indicate Potsdam location in Figure 1.

**Answer**: To establish the accuracy of the method, we added an additional figure comparing hourly observed meteorology with the disaggregated meteorology randomly generated from the same daily means.

Moreover, we added the following text to section 2.5 in the methods:

25  "This method accurately reproduced the complexity of the observed wind speed dynamics: It produced realistic behavior of day-to-day wind speed (Fig. 3a), as well as the diurnal variation of mean wind speed and associated variance (Fig. 3b, c), and also the seasonal change of this diurnal variation (Fig. 3b, c), while still preserving the given daily mean wind speeds."

Note that the observed weather data are from the station at Menz, not Potsdam as we mistakenly stated in the original. Thus, we changed Potsdam to Menz throughout the manuscript. Menz is 5 km from Stechlinsee and the small distance could not be

30  resolved in Fig 1. This we described the location in the text instead:

"…from the Menz weather station 5 km from Lake Stechlin (Fig. 1)."

**Comment:** Temporal downscaling of humidity is performed linearly. Is it relative humidity that is considered as input to FLake? Usually specific humidity is used. Please clarify.

**Answer**: We stated in the manuscript (section 2.5 in the methods) that Flake uses specific humidity.

**Comment:** In the presentation of FLake model runs, it would have been helpful to clearly explain which time step was used: sub-daily variables are constructed but then is the atmospheric forcing 6-hourly, 3-hourly, etc.?

5   **Answer**: The sentence in section 2.5, methods, now reads:

"These variables were available at daily resolution in the climate projections, and were downscaled for model simulations to 6-hourly resolution with the same daily mean to account for diurnal forcing."

**Comment:** Page 9 line 9: ensemble; Page 12 line 25: transparency; Page 12 line 28: Heiligensee

10   **Answer**: Thanks, we corrected these typos.

**Revisions in response to referee #2**

15   We thank referee #2 for providing constructive comments.

General comments

**Comment:** Paper addresses relevant scientific questions within the scope of HESS, namely internal physical mechanisms determining the response of lakes to a future warmer climate. It presents new analysis of northern temperate lakes variables in

20   a projected moderate climate warming scenario (Radiative Concentration Pathway 4.5, RCP4.5). In addition, all lakes used in this study have different combination of morphology and mixing regime, yet they all are situated rather close to each other, what makes analysis even more interesting and relevant. Paper reaches substantial conclusions on lake vertical mixing, ice formation dates and water transparency behaviour according to the projected climate change. Methods and assumptions are valid and rather clearly outlined, the only clarification is needed for light extinction constant for Arendsee. Paper results are

25   sufficient to support the interpretations and conclusions presented. Model experiment description and explanations of result calculation methodology are sufficiently complete and precise to allow their reproduction by fellow scientists (good traceability of results). Also, authors give possibility to download initial data or model output. They give proper credit to related work and clearly indicate their own new/original contribution to the analysis of lake main variables in future warming climate and indicate each authors input. Paper title clearly reflects the contents of the manuscript, abstract provides a concise

30   and complete summary of the research done.

Overall presentation of the paper is well structured and clear, language fluent and precise, all mathematical formulae, symbols, abbreviations, and units are correctly defined and used, number and quality of references are appropriate.

**Answer:** Thank you for the overview and highlighting the strengths of the manuscript. Our response to the extinction coefficient is given below.

Specific comments

**Comment:** 1. p4, l13 - could you add some explanation how constant 2.17 was derived?

**Answer**: We added the expanded the description in section 2.2 in the methods, which now reads:

"Using regression, we related light extinction to parallel measurements of Secchi depth using the relationship $\gamma = c / h_{secchi}$ (Poole and Atkins, 1929). We determined the constant $c$ to be 2.05±0.04 (mean ± s.e, n=300) for Müggelsee, 2.13±0.10 (n=52) for Heiligensee (Shatwell et al., 2016), and 2.33±0.08 (n=57) for Stechlinsee. In the absence of direct measurements in Arendsee, light extinction was estimated from Secchi depth measurements as $\gamma = 2.17 / h_{secchi}$, where the constant $c = 2.17$ was simply the mean of the estimates from the other three lakes."

**Comment:** 2. p6, l5 - could you specify on the technique used to detect lake variable changes for rather small lakes (lake water surface area vary between 0.3- 7.3 km2) if simulations were performed at a horizontal resolution 0.44_? Or this is only atmospheric forcing resolution?

**Answer**: This was the resolution of the atmospheric forcing. For clarification, we stated at the end of section 2.4 in the methods:

"All simulations were performed at a horizontal resolution of atmospheric forcing data of 0.44°."

**Comment:** 3. p6, l17 - what period of data was used to empirically determine the value?

**Answer**: We added the period to the text in section 2.5 of the methods:

"The constant $m$ was empirically determined to be 1.3 based on high resolution weather data (1 Nov 1991 to 24 Aug 2004) from the Menz weather station 5 km from Lake Stechlin (Fig. 1)."

**Comment:** 4. p27, Fig.5 - Stechlinsee and Arendsee patterns look very similar, horizontal and vertical grids would help to better see if any difference is present.

**Answer**: we added gridlines to the plots in Fig 5 and slightly expanded the vertical scale.

**Comment:** 5. p28, Fig.6 - mixed layer depth values especially for Stechlinsee and Arendsee are not visible (as well as winter and autumn periods for all 4 lakes), as it was mentioned that there are 58% and 75% respectively that these lakes are ice-free is it possible to show values on the graph (extending y and x axis)? Or an explanation why it shouldn't be done?

**Answer**: We did not alter the figure as suggested by the reviewer. We explained our reasoning in our previous response to reviewer 2.

**Comment:** 6. p30, Fig.8 - could you explain an interesting behaviour pattern of Heiligensee in annual mean temperature graph?

**Answer**: We added a potential explanation for this behavior in section 4.5 of the discussion:

"We assume that the interesting behavior of Heiligensee in Fig. 10j is probably due to how the mixed layer (if present) interacts with the lake bottom, making the heat budget of this lake especially sensitive to the mixing regime."

**Comment:** 7. p32, Table1 - could extra line with ice duration in days be added?

5 **Answer**: We added ice duration statistics for the two lakes for which we had ice data (Müggelsee and Stechlinsee).

**Comment:** 8. p33, Table2 - some correction with table rows is needed.

**Answer**: Thanks for the info, this seems to be a formatting issue which should disappear during typesetting.

[revised manuscript text omitted]